# SVG: 3D STEREOSCOPIC VIDEO GENERATION VIA DENOISING FRAME MATRIX

Peng Dai[1,2]*    Feitong Tan[1]†    Qiangeng Xu[1]†    David Futschik[1]    Ruofei Du[1]
Sean Fanello[1]    Xiaojuan Qi[2]    Yinda Zhang[1]
[1]Google    [2]The University of Hong Kong

## ABSTRACT

Video generation models have demonstrated great capability of producing impressive monocular videos, however, the generation of 3D stereoscopic video remains under-explored. We propose a pose-free and training-free approach for generating 3D stereoscopic videos using an off-the-shelf monocular video generation model. Our method warps a generated monocular video into camera views on stereoscopic baseline using estimated video depth, and employs a novel *frame matrix* video inpainting framework. The framework leverages the video generation model to inpaint frames observed from different timestamps and views. This effective approach generates consistent and semantically coherent stereoscopic videos without scene optimization or model fine-tuning. Moreover, we develop a disocclusion boundary re-injection scheme that further improves the quality of video inpainting by alleviating the negative effects propagated from disoccluded areas in the latent space. We validate the efficacy of our proposed method by conducting experiments on videos from various generative models, including Sora (Brooks et al., 2024), Lumiere (Bar-Tal et al., 2024), WALT (Gupta et al., 2023), and Zeroscope (Wang et al., 2023a). The experiments demonstrate that our method has a significant improvement over previous methods. Project page at https://daipengwa.github.io/SVG_ProjectPage/

## 1 INTRODUCTION

As VR/AR technology advances, the demand for creating stereoscopic content and delivering immersive 3D experiences to users continues to grow. Due to visual sensitivity, binocular stereoscopic content should feature flawless 3D and semantic consistency between both eye views, as well as seamless temporal consistency across frames. While monocular video generation models have been extensively researched and methods are now capable of synthesizing high-fidelity videos that adhere to complex text prompts (Brooks et al., 2024), there has not been much progress in the realm of generating 3D stereoscopic videos at the scene level. One reason for this gap lies in the substantial amount of monocular video data that is readily available, contrasted with the scarcity of stereo video data for training models to generate stereoscopic videos directly.

An emergent solution is to convert generated monocular videos into stereoscopic videos using novel view synthesis (Li et al., 2023; Liu et al., 2023b). However, these methods usually overly rely on camera pose estimation, which is a challenging task on its own either using SFM (Schönberger & Frahm, 2016) or joint optimization (Liu et al., 2023b), and as a result tend to be unstable, particularly in dynamic scenes where cameras experience subtle motions or when the content is dominated by dynamic objects with temporally varying appearances, both of which are prevalent in generated videos. Consequently, these methods fail in optimizing 3D scenes and offer low-quality solutions to the task (see Fig. 3). Moreover, these approaches are based on reconstruction, lacking the generative ability to hallucinate occluded regions in the novel views that do not appear in any of the remaining video frames.

In this paper, we propose an alternative pose-free and training-free framework, for the sake of robustness and generalization capability, that operates solely by exploiting inference of an off-the-

---

*Work done when Peng Dai was an intern at Google

†Equal contribution

shelf video generation model (Wang et al., 2023a) to generate high-quality 3D stereoscopic videos. Our initial attempt follows a typical 2D to 3D image uplifting methodology (Höllein et al., 2023) and extends it into the video domain. Specifically, we first generate a monocular video as the left view, which is then warped into the right view using estimated monocular depths (Yang et al., 2024), where we apply temporal-spatial smoothing to improve the consistency of the estimated depths. Subsequently, we leverage an off-the-shelf video generation model's ability (Wang et al., 2023a) to generate natural videos, by adding noise and denoising the warped video frames to inpaint the disoccluded regions, inspired by diffusion-based image inpainting (Avrahami et al., 2023).

However, this naive pipeline does not produce appealing results: inpainting the right-view video frames independently, without referencing the left view, typically generates semantically mismatched content. To address this problem, we propose a novel representation, called the *frame matrix*, which contains frame sequences observed from a number of viewpoints evenly distributed along the baseline between two eyes. The frame sequences along the view direction (rows of the matrix) form videos with camera motion, while the frame sequences along the time direction (columns of the matrix) form videos with scene motions (see Fig. 1 second column). Since the video diffusion model has video prior for both scene and camera motions, we propose to jointly update the entire *frame matrix* from both directions. In each denoising step, we use resample techniques (Lugmayr et al., 2022) by alternatively denoising frame sequences along the view and the time directions. Finally, we obtain a semantically consistent and temporally smooth 3D stereoscopic video by taking the leftmost and the rightmost frame sequences to represent the left-eye view and the right-eye view, respectively.

Furthermore, we note that the inevitable resolution downsampling operation in most video generation models with latent encoding (Brooks et al., 2024; Bar-Tal et al., 2024; Wang et al., 2023a; Gupta et al., 2023) is detrimental to the video inpainting task. During encoding, the dark pixels created by disocclusion can degrade the features near the disocclusion boundary, leading to undesirable artifacts (see Fig. 5). Instead of following the inpainting scheme proposed in previous work (Avrahami et al., 2023), which encodes the latent feature only once, we iteratively update both the disoccluded regions in the image space and the latent feature map with generated content during the diffusion process. This approach re-injects the generated content into the disocclusion boundary, which mitigates the negative impact of disoccluded regions in downsampling and effectively prevents the artifacts.

To validate the efficacy of our proposed method, we generate stereoscopic videos from monocular videos generated by Sora, Lumiere, WALT, and Zeroscope. Both qualitative and quantitative evaluations suggest that our approach outperforms other baselines in 3D stereoscopic video generation. Our contributions are summarized as follows:

- We design a novel pipeline to generate 3D stereoscopic videos. Unlike previous work, our method does not need camera pose estimation or fine-tuning on specific datasets.
- We propose a novel *frame matrix* representation that regularizes the diffusion-based video inpainting to generate semantically consistent and temporally smooth content.
- We propose a re-injection scheme that drastically reduces the negative influence of disoccluded regions in latent space and produces high-quality results.
- We conduct comprehensive experiments that show the superiority of our approach over previous methods for 3D stereoscopic video generation.

## 2 RELATED WORK

**Video Generation.** Video generation (Wang et al., 2023a; Brooks et al., 2024; Bar-Tal et al., 2024; Gupta et al., 2023; Harvey et al., 2022; Ho et al., 2022a;b; Singer et al., 2022) has achieved tremendous progress since the advent of the diffusion model (Ho et al., 2020). Taking into account the dataset requirements and scarcity of tagged videos, a prominent approach for video generation is to extend pre-trained image generation models (Rombach et al., 2022; Saharia et al., 2022; Ramesh et al., 2022) by inserting additional temporal layers and then fine-tuning them on video data (Guo et al., 2023; Blattmann et al., 2023; Wu et al., 2023b). To further improve the compute efficiency and enable long clip processing, WALT (Gupta et al., 2023) and Lumiere (Bar-Tal et al., 2024) proposed to compress the video in both the temporal and spatial dimensions. More recently, Sora (Brooks et al., 2024) adopted a transformer diffusion architecture (Peebles & Xie, 2023) and was trained on large-scale video datasets to produce impressive video generation results. Different from previous

video generation models focusing on producing higher-quality and longer monocular videos, our method orthogonally explores the possibility of leveraging pre-trained video generation models for stereoscopic 3D video generation.

**Novel View Synthesis.** Great progress has been made for novel view synthesis in both static and dynamic scenes (Mildenhall et al., 2021; Yoon et al., 2020; Li et al., 2022b; Kerbl et al., 2023; Müller et al., 2022; Tucker & Snavely, 2020; Han et al., 2022; Li et al., 2022a; Wang et al., 2023b).Tucker & Snavely (2020) convert a single image into a multi-plane representation for view synthesis. Mildenhall et al. (2021) proposed to encode the static scene into neural radiance fields (NeRF), which were then used for novel view synthesis through volume rendering. For more challenging scenes with dynamic content, follow-up works additionally optimized a deformation field (Park et al., 2021a; Huang et al., 2023; Park et al., 2021b) or scene flow fields (Li et al., 2021b) to handle the motion of dynamic objects. Instead of encoding the scene into a NeRF, DynIBaR (Li et al., 2023) leveraged nearby frames for rendering novel view images, and dynamic objects were handled by optimized motion fields. Different from methods requiring pre-computed camera poses, RoDynRF (Liu et al., 2023b) jointly optimized the NeRF and camera poses from scratch. Concurrently, FVS (Lee et al., 2023) achieves novel view video synthesis using a plane-based scene representation. Although these approaches produce high-quality renderings, they are limited to scenes where the camera pose can be accurately estimated and have limited synthesis capability. In contrast, our method explicitly avoids having to estimate camera poses and possesses the ability to hallucinate unseen content.

**3D Content Creation and Inpainting.** Automated 3D content creation (Höllein et al., 2023; Dai et al., 2024; Gao et al., 2024; Yu et al., 2023; Chen et al., 2024) is another related area, with emerging approaches such as inpainting (Ho et al., 2022a) or multi-view generators (Liu et al., 2023a; Wang et al., 2024; Zuo et al., 2024). Recently, Text2Room (Höllein et al., 2023) proposed creating a 3D room by warping an image into novel views and using a text-guided inpainter to deal with disocclusions. WonderJourney (Yu et al., 2023) made this process automatic by including a large language model in the loop. Similar to creating static scenes, we could use pretrained video inpainter (Zhou et al., 2023; Li et al., 2022c) for dynamic 3D content creation, however, these models suffer from generalization problems in creating high-quality, consistent 3D content. Lastly, Deep3D (Xie et al., 2016) is trained using 3D movies, with the goal of converting 2D videos into stereoscopic videos. However, the training data is not publicly available and it lacks the flexibility to modify videos for creative purposes, such as different stereo baselines. In this paper, we explore the possibilities of using video generation models for 3D video creation without training on specific, hard-to-obtain datasets.

## 3 STEREOSCOPIC VIDEO GENERATION

Conditioned on a text prompt or a single image $c$, our method aims to generate 3D stereoscopic video $\{\mathbf{X}_l, \mathbf{X}_r\}$, consisting of two monocular sequences. The most straightforward way is to use a diffusion-based generation model $\mathcal{G}$:

$$\{\mathbf{X}_l, \mathbf{X}_r\} = \mathcal{G}\left(\{\epsilon_t \mid t = 1, ..., T\}, c\right), \tag{1}$$

where $\epsilon_t \sim \mathcal{N}(\mathbf{0}, \mathbf{I})$ is the sampled noise at step $t$. The generated stereoscopic videos should possess the following characteristics: First, the appearance and semantics between the left eye view $\mathbf{X}_l$ and right eye view $\mathbf{X}_r$ should be consistent and be temporally stable. Second, the stereo effect should be prominent and immersive. Last, the generated content should be diverse and controllable with the given conditioning.

However, training a $\mathcal{G}$ that can directly generate stereo videos $\{\mathbf{X}_l, \mathbf{X}_r\}$ with the desired properties requires a vast dataset of stereo videos with diverse content. Due to the scarcity of such data, we propose a training-free approach that relies on an off-the-shelf depth estimator (Yang et al., 2024) and a diffusion-based monocular video generation model $\mathcal{G}$ such as Zeroscope (Wang et al., 2023a). We first generate a monocular video for one eye using a video diffusion model (Eq. 2), then obtain the other video view by conditioning on the first video. To automatically preserve 3D consistency, we implement this conditioning by estimating depth $\mathbf{d}_l$ for the left video and warp its content to obtain the right view sequence $\mathbf{X}_{l \to r}$ with disocclusion masks $\mathbf{M}_r$ (Eq. 3) according to the stereoscopic baseline (Wang et al., 2023b; Han et al., 2022). Then, we use $\mathcal{G}$ again to inpaint the disoccluded parts by denoising inpainting process (Avrahami et al., 2023; Lugmayr et al., 2022) (Eq. 4), obtaining the

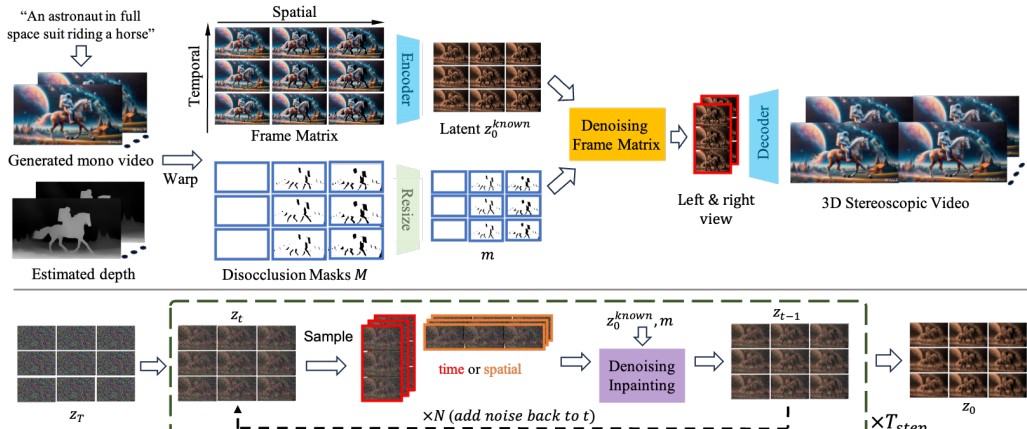

Figure 1: **Overview** – **Top:** Given a text prompt, our method first uses a video generation model to generate a monocular video, which is warped (using estimated depth) into pre-defined camera views to form a *frame matrix* with disocclusion masks $M$. Then, the disoccluded regions are inpainted by denoising the frame sequences within the *frame matrix*. After denoising, we select the leftmost and the rightmost columns and decode them to obtain a 3D stereoscopic video. **Bottom:** Details of denoising *frame matrix*. We initialize the latent matrix $\mathbf{z}_T$ as a random noise map. For each noise level, we extend the resampling mechanism (Karnewar et al., 2023; Lugmayr et al., 2022) to alternatively denoise temporal (column) sequences and spatial (row) sequences $N$ times. Each time, row or column sequences are denoised and inpainted (see Fig.2). By denoising along both spatial and temporal directions, we obtain an inpainted latent $\mathbf{z}_0$ which can be decoded into temporally smooth and semantically consistent sequences.

other eye view video $\mathbf{X}_r$.

$$\mathbf{X}_l = \mathcal{G}\left(\{\epsilon_t \mid t = 1, ..., T\}, c\right), \tag{2}$$

$$\mathbf{X}_{l \to r}, \mathbf{M}_r = \mathrm{Warp}_{l \to r}(\mathbf{X}_l, \mathbf{d}_l), \tag{3}$$

$$\mathbf{X}_r = \mathcal{G}\left(\{\epsilon_t \mid t = 1, ..., T\}, c, \mathbf{X}_{l \to r}, \mathbf{M}_r\right). \tag{4}$$

In Sec. 3.1, we describe the video depth warping. In Sec. 3.2, we introduce the *frame matrix* representation for the video inpainting. Our denoising frame matrix drastically improves the semantic similarity between $\mathbf{X}_l$ and $\mathbf{X}_r$ and helps preserve temporal smoothness. Last but not least, a disocclusion boundary re-injection mechanism is introduced to further improve the inpainting quality in Sec. 3.3. An overview of our method is displayed in Fig. 1.

## 3.1 MONOCULAR VIDEO DEPTH WARPING

The depth estimation model (Yang et al., 2024) is applied to predict all frames' depth values, which will be smoothed to produce more consistent video depths. Specifically, we utilize the estimated optic flows (Teed & Deng, 2020) to align consecutive depth frames. The outliers in predicted depths will be suppressed by convolving with a Gaussian kernel along the time axis. After obtaining RGB-D frames, we can warp them into target camera views where disoccluded regions appear. In addition, the warped images usually contain isolated pixels, and the foreground and background are entangled, which jeopardizes video quality (Dai et al., 2020). To handle these problems, we follow Dai et al. (2020) to project points into multi-plane images (Zhou et al., 2018), then remove isolated pixels and cracks and finally obtain a noisy-points-free image. (See supplemental material Sec. C for details).

## 3.2 VIDEO INPAINTING WITH FRAME MATRIX

The inpainting pipeline plays a key role in ensuring spatial/semantic and temporal consistency. While image inpainting approaches (Avrahami et al., 2023; Lugmayr et al., 2022) provide a reasonable baseline, the results lack temporal and spatial stability. Therefore, we introduce a Frame Matrix representation, which addresses both issues.

**Single Video Denoising Inpainting** Inspired by RePaint (Lugmayr et al., 2022), we extend the diffusion-based image inpainting to video inpainting. We use the video generation model $\mathcal{G}$ (i.e.,

Zeroscope (Wang et al., 2023a)) as our inpainting tool, which is a latent diffusion model consisting of a VAE encoder $\mathcal{E}$, a decoder $\mathcal{D}$ and a latent denoiser $\{\epsilon_\theta, \Sigma_\theta\}$. First, the warped video is fed into the VAE encoder to obtain video latent features $\mathbf{z}_0^{\text{known}} = \mathcal{E}(\mathbf{X}_{l \to r})$. Then, we resize the image disocclusion masks $\mathbf{M}_r$ to the resolution of the latent and obtain latent disocclusion masks $\mathbf{m}$. During the denoising process, we start from a random noisy latent map $\mathbf{z}_T \sim \mathcal{N}(\mathbf{0}, \mathbf{I})$. For each subsequent step $t$, we sample a new intermediate noisy latent map from $\mathbf{z}_0$ (Eq. 5), denoises the latent map from the last step $\mathbf{z}_t$ (Eq. 6) and combine them with $\mathbf{m}$ to obtain the $\mathbf{z}_{t-1}$ (Eq. 7). We visualize the following steps in Fig. 2 (b):

$$\mathbf{z}_{t-1}^{\text{known}} \sim \mathcal{N}\left(\sqrt{\bar{\alpha}_t}\mathbf{z}_0^{\text{known}}, (1-\bar{\alpha}_t)\mathbf{I}\right), \tag{5}$$

$$\mathbf{z}_{t-1}^{\text{unknown}} \sim \mathcal{N}\left(\frac{1}{\sqrt{1-\beta_t}}\left(\mathbf{z}_t - \frac{\beta_t}{\sqrt{1-\bar{\alpha}_t}}\epsilon_\theta(\mathbf{z}_t, c, t)\right), \Sigma_\theta(\mathbf{z}_t, c, t)\right), \tag{6}$$

$$\mathbf{z}_{t-1} = m \odot \mathbf{z}_{t-1}^{\text{known}} + (1-m) \odot \mathbf{z}_{t-1}^{\text{unknown}}, \tag{7}$$

where $\bar{\alpha}_t$ and $\beta_t$ denote the total noise variance and one step noise variance at $t$, respectively; $\epsilon_\theta(\mathbf{z}_t, c, t)$ and $\Sigma_\theta(\mathbf{z}_t, c, t)$ are predicted noise and variance for noisy latent map at $t-1$ step. Finally, we can obtain the inpainted right view sequence $X_r$ by decoding the denoised latent $X_r = \mathcal{D}(\mathbf{z}_0)$.

By applying the above video inpainting scheme for the right view, we implement Eq. 4 and successfully hallucinate the disoccluded (unknown) regions while preserving the unoccluded (known) regions. The video diffusion model also ensures temporal smoothness. However, the inpainted content on the right view usually lacks semantic consistency w.r.t. the left view, as shown in the third column of Fig. 4. This is because we only condition on the left view by depth warping, while dropping the conditioning during inpainting.

**Frame Matrix Representation.** We propose a novel representation–*frame matrix*, which targets consistent dynamic content generation across space and time. As shown in Fig. 1 top, it is a matrix consisting of multiple frames, where each row represents frames observed from different camera poses at the same time stamp, and each column is a video recorded by a fixed camera at different time stamps. Consequently, the frame matrix can be defined as:

$$\mathbf{X} \equiv \left[\begin{array}{ccc} | & & | \\ \mathbf{x}_{(:,0)} & \cdots & \mathbf{x}_{(:,V)} \\ | & & | \end{array}\right] \equiv \left[\begin{array}{ccc} - & \mathbf{x}_{(0,:)} & - \\ & \vdots & \\ - & \mathbf{x}_{(S,:)} & - \end{array}\right]$$

where $S$ and $V$ are the largest indices of time steps and views, respectively. A view sequence (row) $\mathbf{X}_{(s,:)}$ forms a video with camera motions, while a time sequence (column) $\mathbf{X}_{(:,v)}$ forms a video with time-varying scene motions. Since the video diffusion model can denoise a sequence to a temporally and semantically consistent video, jointly denoise the rows and columns can ensure consistency spatially and temporally. Finally, we can obtain a 3D stereoscopic video by taking the leftmost and the rightmost time sequences $\mathbf{X}_{(:,0)}, \mathbf{X}_{(:,V)}$.

**Constructing Frame Matrix.** We evenly add $V$ camera views distributed along the baseline between the two eyes with the same orientation of the reference view. Then, we warp the reference video (the $0th$ column) based on depth (Sec. 3.1) into these views and obtain $\mathbf{X}_{warp} \equiv \left[\mathbf{X}_{(:,0)}, \mathbf{X}_{(:,0 \to 1)}, ..., \mathbf{X}_{(:,0 \to V)}\right]$ with a disocclusion masks matrix $\mathbf{M}$.

**Denoising Frame Matrix.** Similar to single video sequence inpainting, we encode frame matrix into a latent frame matrix $\mathbf{z}_0^{\text{known}} = \mathcal{E}(\mathbf{X}_{warp})$, and resize $\mathbf{M}$ to obtain latent disocclusion map $\mathbf{m}$. We also initialize $\mathbf{z}_T \sim \mathcal{N}(\mathbf{0}, \mathbf{I})$. As shown in Fig.1 (Bottom), for each noise level, we extend the resampling mechanism (Lugmayr et al., 2022) to alternatively denoise column sequences and row sequences $N$ times. Each time, row or column sequences are denoised following Eq. 5-7 and we add back noise between every resampling iteration:

$$\mathbf{z}_t \sim \mathcal{N}(\sqrt{1-\beta_{t-1}}\mathbf{z}_{t-1}, \beta_{t-1}\mathbf{I}). \tag{8}$$

Please refer to Sec.B in the supplemental material. By denoising along these two directions alternatively, the spatial and temporal sequences will proceed toward a harmonic state.

**High-level motivation of Frame Matrix.** In practice, 3D stereoscopic videos can be produced by recording with two cameras (time-direction videos). Since both cameras are capturing the same

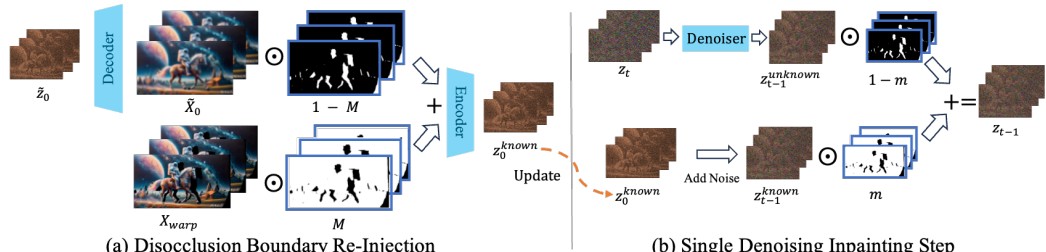

(a) Disocclusion Boundary Re-Injection        (b) Single Denoising Inpainting Step

Figure 2: **Denoising Inpainting**. This figure visualizes the operations in the purple box of Fig.1. (a) We re-inject the generated content from a denoised latent $\widetilde{\mathbf{z}}_0$ to update $\mathbf{z}_0^{known}$ and reduce its feature corruption on the disocclusion boundary. (b) A noisy latent $\mathbf{z}_t$ is denoised to $\mathbf{z}_{t-1}^{unknown}$. We take its disoccluded region and combine it with the unoccluded region of $\mathbf{z}_0^{known}$.

scene, gradually moving the left camera toward the right camera also results in a coherent video (space-direction video). Likewise, it is essential to consider both the time and spatial directions when generating the 3D stereoscopic video to ensure that both perspectives represent the same scene. Additionally, compared to direct inpainting a large region all at once, gradually expanding the inpainting area (spatial direction) tends to be easier and can lead to more stable and plausible results. More analysis is in supplementary material Sec. D.

### 3.3 DISOCCLUSION BOUNDARY RE-INJECTION

Since most video generation models are using latent diffusion, the disoccluded dark regions of $\mathbf{X}_{warp}$ will be propagated beyond the latent mask $\mathbf{m}$ during VAE encoding (*e.g.*, Zeroscope downsamples by $8\times$), leading to defective latent features on $\mathbf{z}_0^{known}$'s disocclusion boundary. This would lead to artifacts in the final results (Fig. 5 left).

We propose to re-inject the denoised information in the disoccluded regions to improve the latents on this boundary. Specifically, we predict the denoised latent features (Ho et al., 2020), which are decoded into a denoised video (Eq. 9). Then, we replace its unoccluded regions with warped pixels to form a video that is faithful to the reference view but with better disocclusion pixels. By encoding this video, we can get a updated $\mathbf{z}_0^{known}$ (Eq. 10) which alleviates corruption on the boundary:

$$\widetilde{\mathbf{X}}_0 = \mathcal{D}(\widetilde{\mathbf{z}}_0), \text{ where } \widetilde{\mathbf{z}}_0 = \frac{1}{\sqrt{\bar{\alpha}_t}}\left(\mathbf{z}_t - \sqrt{1 - \bar{\alpha}_t}\epsilon_\theta(\mathbf{z}_t, c, t)\right), \tag{9}$$

$$\mathbf{z}_0^{known} = \mathcal{E}\left(\mathbf{M} \odot \mathbf{X}_{warp} + (1 - \mathbf{M}) \odot \widetilde{\mathbf{X}}_0\right). \tag{10}$$

After this, this improved $\mathbf{z}_0^{known}$ can be used in Eq. 5 for the next iteration.

## 4 EXPERIMENTS

**Datasets.** To validate the effectiveness of our method, we conduct experiments using a variety of recent video generation models, including Sora (Brooks et al., 2024), Lumiere (Bar-Tal et al., 2024), WALT (Gupta et al., 2023), and Zeroscope (Wang et al., 2023a). These models produce diverse left videos from a wide range of input text prompts, covering subjects such as humans, animals, buildings, and imaginary content.

**Implementation Details.** To ensure the stereo effect appears realistic, we normalize the up-to-scale depth values predicted by the depth estimation model (Yang et al., 2024) to a range of (1, 10) and set the baseline between left and right views to 0.08. The frame matrix is constructed by evenly placing 8 cameras between the left and right views, with each camera corresponding to a warped video sequence. Due to the limitations of the Zeroscope model, we currently conduct experiments on video sequences with 16 frames. Following the approach of RePaint (Lugmayr et al., 2022), we employ DDPM (Ho et al., 2020) as our denoising scheduler with 1000 total time steps $T$ and 50 denoising steps, resulting in 20 time step jumps per denoising step. During the initial 25 denoising steps (50 to 25), we resample 8 times at each step to establish a reasonable structure in disoccluded regions. For the remaining steps, we reduce resampling to 4 times and denoise only the right view for improved efficiency while generating stereoscopic videos. Currently, our implementation runs on an A6000 GPU and takes $\sim$8 minutes to generate stereoscopic video using only $\sim$10GB of RAM.

**Baselines.** We compare our method with two families of approaches: video inpainting, and novel view synthesis from a monocular video. For video inpainting approaches, we generate the right view in the same manner as our method using depth-guided warping. We then apply state-of-the-art methods ProPainter (Zhou et al., 2023) and E2FGVI (Li et al., 2022c) to inpaint the right views. For novel view synthesis methods, we compare our results with RoDynRF (Liu et al., 2023b) and DynIBaR (Li et al., 2023), which optimize scene representations relying on camera poses. To ensure a fair comparison, given the differing 3D scales between their reconstructed scenes and our estimated depth, we select the baseline for rendering the right view by matching the median disparity of foreground regions in the resulting disparity map to that of our methods. We are also aware of approaches trained on dedicated datasets that directly produce the right view given the left view like Deep3D (Xie et al., 2016). However, it does not generalize well to the generated video, especially those in non-realistic styles, and the comparison could be found in the supplemental material.

## 4.1 QUALITATIVE RESULTS

**Qualitative Comparisons.** We show qualitative comparisons in Fig. 3. Previous video inpainting methods suffer from a common problem – the generated content in disoccluded regions is blurry, such as the knight's arm, horse's tail, and corgi's face, presumably because that these methods are trained on limited datasets. On the other hand, novel view synthesis methods suffer from unstable camera pose estimation (*e.g.*, DynIBaR fails on some videos). Though good at reconstructing visible content from the monocular video, they are typically poor at synthesizing novel contents in the disoccluded regions that are not observed in any frames (*e.g.*, ghost effect near the boundary in the RoDynRF result on the corgi example). In contrast, our approach takes advantage of the generative capability of video diffusion models trained on massive scale datasets and does not require camera poses of the input video as inputs, thereby generating high-quality content in various types of scenarios (last row of Fig. 3) and consistently outperforms baseline methods. Additionally, we visualize the stereo effects of different methods on the corgi case using a stereo depth estimator (Li et al., 2021a), which predicts disparity values from the stereo images. As shown in Fig. 10, RoDynRF and DynIBaR exhibit less depth variation, indicating weaker stereo effects. This occurs when the camera is wrong and the training process overfits the training views, resulting in a sub-optimal 3D representation.

## 4.2 QUANTITATIVE RESULTS

In this part, we show quantitative comparisons with other baselines. We primarily rely on a dedicatedly designed user study to evaluate the quality of generated stereoscopic video on various quality axes. We also provide an objective metric to measure the semantic similarity between the left and right views using pre-trained CLIP models.

**Human Perception.** To assess the perceived visual quality, we conducted a user study with 20 participants (9 female, age $\mu = 33, \sigma = 6.2$). On a VR headset, each participant viewed and evaluated five generated videos (out of 20 in total) by all five methods on stereo effect, temporal consistency, image quality, and overall experience using a 7-point Likert scale (Likert, 1932). A total of 435 evaluations (DynIBaR failed to generate 13 videos) were counterbalanced and randomly shuffled. We also included a training session to eliminate novelty effects. Results are summarized in Table 1, with details in the supplemental material. Our method outperforms other baselines on measured metrics.

| | E2FGVI | ProPainter | RoDynRF | DynIBaR | Ours |
|---|---|---|---|---|---|
| Stereo Effect ↑ | 4.79 (1.08) | 4.81 (1.13) | 2.97 (1.34) | 1.86 (1.25) | **5.24** (0.94) |
| Temporal Consistency ↑ | 4.74 (1.33) | 4.74 (1.22) | 3.35 (1.66) | 1.89 (1.33) | **5.15** (1.22) |
| Image Quality ↑ | 4.42 (1.27) | 4.38 (1.28) | 2.84 (1.60) | 1.67 (1.07) | **5.12** (1.33) |
| Overall Experience ↑ | 4.67 (1.04) | 4.66 (1.09) | 2.92 (1.43) | 1.72 (1.06) | **5.35** (0.99) |

Table 1: **Human perception.** This table reports the results of human perception experiments as mean (std). Our method outperforms other baselines on all metrics. Kruskal-Wallis tests (Kruskal & Wallis, 1952) reveal significant effects of group on all metrics ($\chi^2 > 13.3, p < 0.001$***). Post-hoc tests using Mann-Whitney tests (Mann & Whitney, 1947) with Bonferroni correction reveal significant effects ($p < 0.05$*, $|r| > 0.1$) for each pairwise comparison, except E2FGVI *vs.* ProPainters yield comparable results.

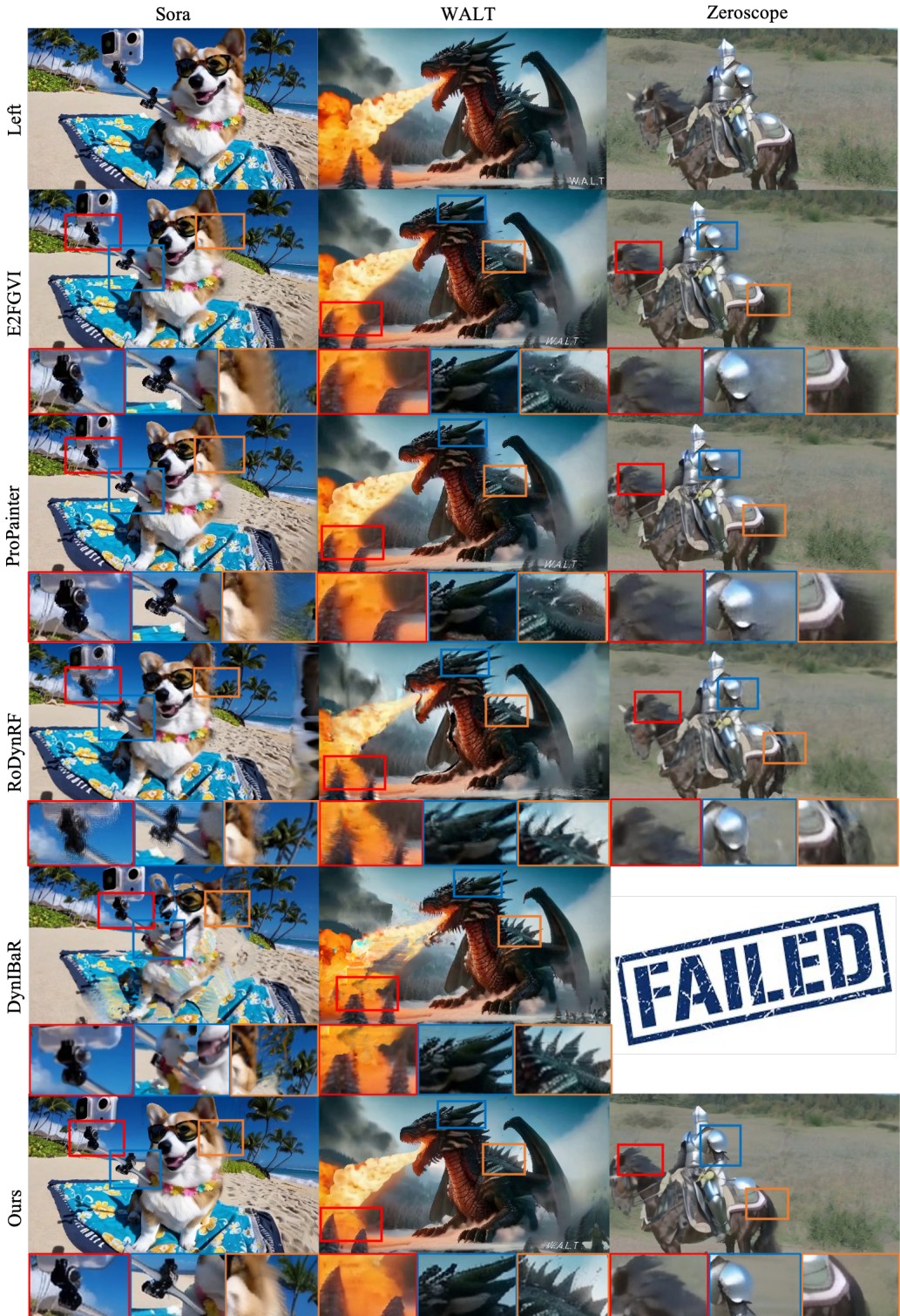

Figure 3: **Qualitative comparisons.** The first row shows left-view images. The video inpainting methods E2FGVI and ProPainter tend to generate blurry content in disoccluded regions, such as knight's arm and corgi's face. RoDynRF lacks the generation ability, thus content on the right side of the corgi case is poor. DynIBaR's results contain artifacts, and it requires camera poses as inputs, which failed in some scenarios. On the contrary, our method takes advantage of video generation models and is pose-free, thus generating high-quality content in different scenarios.

**Semantic Consistency.** We additionally check the semantic consistency between the left and the right view. We use pre-trained CLIP model (Radford et al., 2021) to extract features for both left views and right views of a stereoscopic video, and then calculate the feature distance following Zhengwentai (2023) to obtain the semantic consistency score. In Table 2, our method attains the best semantic consistency (96.44) over other baselines.

**Quality Assessment.** We measure the quality of generated video using aesthetic score (Schuhmann et al., 2022), DOVER (Wu et al., 2023a), and FVD (Unterthiner et al., 2019), which are in line with human preferences. In Table 2, our approach achieves the best performance (5.27, 0.584, and 599), aligning with the conclusion from the user study experiment in Table 1.

## 4.3 ABLATION STUDIES

**Effect of Frame Matrix.** In Fig. 4, we showcase that using frame matrix benefits semantic consistency between the left and right views. Without using frame matrix, the disoccluded regions in warped images can be inpainted with unconstrained contents, which are likely to be inconsistent with the left view given impressive generative capability of the diffusion model, such as the hair of the man and the head of the horse. This is also revealed in Table 2, where CLIP Score drops from 96.44 to 95.81 when disabling frame matrix. Thanks to constraints from other frames within the frame matrix, our method generates both reasonable foreground and background contents in the disoccluded regions. More studies of frame matrix are included in Sec.F and Sec. G of the supplemental material.

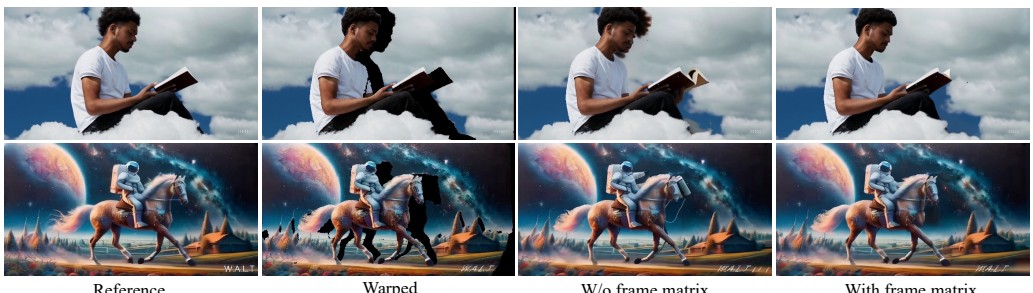

Reference         Warped         W/o frame matrix         With frame matrix

Figure 4: **Semantically consistent content generation.** The reference frames are warped into the target view with disoccluded regions set to be black. Without using frame matrix, the generated content does not match the reference, such as the book and the face of horse. With frame matrix, the inpainted contents are more semantically reasonable.

**Effects of Disocclusion Boundary Re-Injection.** In Fig. 5, we demonstrate the importance of updating unoccluded latent features for high-quality results. Without this update, the disoccluded region is inpainted with unnatural textures that don't blend well with the surrounding content. With the update, the inpainted content blends seamlessly. This is reflected quantitatively in Table 2, where the aesthetic score drops from 5.27 to 5.18 when unoccluded feature updates are discarded.

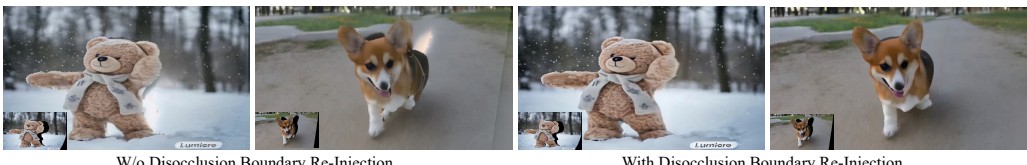

W/o Disocclusion Boundary Re-Injection         With Disocclusion Boundary Re-Injection

Figure 5: **Disocclusion Boundary Re-injection.** Without disocclusion boundary re-injection, the inpainted images usually contain artifacts. Bottom-left corner shows the warped image.

**Effects of Handling Isolated Pixels and Cracks.** In Fig. 6 left, obvious artifacts are in warped images, such as isolated points and cracks where the foreground ear is entangled with the background gray road, and these artifacts remain in the final generated results. After applying warping designs in Sec. C, our results are clean, handling warping-related artifacts as shown in Fig. 6 right side.

**Different Stereo Baselines.** Fig. 7 shows increasing the stereo baseline makes inpainting harder and degrades stereoscopic video quality, as reflected by CLIP score. Our method is resilient to

| Method | E2FGVI | ProPainter | RoDynRF | DynIBaR | Ours - FM | Ours - DBR | Ours |
|--------|--------|-----------|---------|---------|-----------|------------|------|
| CLIP ↑ | 94.34 | 95.29 | 96.03 | 93.24 | 95.81 | 95.60 | **96.44** |
| Aesthetic ↑ | 5.06 | 5.07 | 4.97 | 4.66 | 5.25 | 5.18 | **5.27** |
| DOVER ↑ | 0.547 | 0.535 | 0.352 | 0.365 | 0.565 | 0.560 | **0.584** |
| FVD ↓ | 638 | 606 | 727 | 1208 | 614 | 699 | **599** |

Table 2: **Quantitative comparisons**. We show the semantic consistency using CLIP feature similarity (Hessel et al., 2021) between the left and right view. Additionally, the quality of generated videos is measured by aesthetic score (Schuhmann et al., 2022), DOVER (Wu et al., 2023a) and FVD (Unterthiner et al., 2019). Our method outperforms previous methods as well as ablated cases.

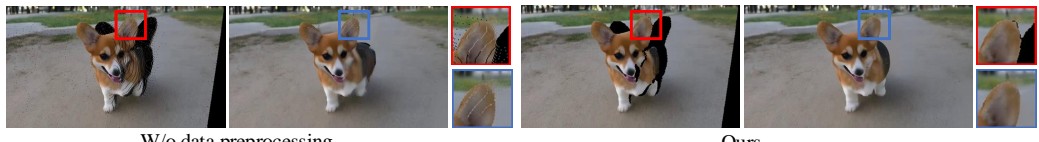
W/o data preprocessing          Ours

Figure 6: **Isolated points and cracks.** Left: without handling isolated points and entangled foreground and background (the gray road can be seen through the cracks) in warped images, these artifacts remain in the final results. Right: our results are clean.

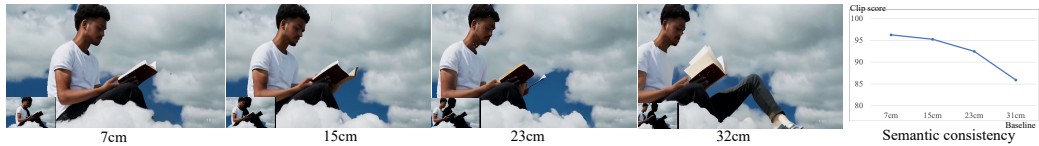
7cm      15cm      23cm      32cm

Figure 7: **Different stereo baselines.** Unnatural artifacts begin to appear as the baseline expands. Our method performs well for stereoscopic video generation where baseline is usually less than 7cm.

larger baselines, failing beyond 20cm (depth normalized to 1.0-10.0m). This range is sufficient for generating 3D stereoscopic video for most people, given typical inter-pupillary distances of 5-7cm.

## 5 LIMITATIONS AND DISCUSSIONS

Although our results demonstrate the possibility of generating 3D stereoscopic videos using pre-trained video diffusion models, challenges remain. For one, we did not study longer videos because the architecture of a typical video diffusion model supports the generation of videos only a few seconds long. One possible solution for long 3D stereoscopic video generation is to use stronger foundational models, such as Sora (Brooks et al., 2024). Alternatively, we could gradually generate longer videos by overlapping frames of shorter videos. Additionally, our method depends on a depth estimation model, which may fail, *e.g.*, when dealing with thin structures or boundaries. In practice, the depth estimation model showcases decent depth consistency after training on large-scale datasets containing images extracted from the same scene and videos, and our flow-based smoothing operation can further mitigate depth perturbations that occasionally arise. To obtain more consistent video depth estimation, fine-tuning video generation models with synthetic video depths is worth exploring. Note that our method is orthogonal to the depth estimation and can benefit from its advancements. Furthermore, our method utilizes auxiliary views to assist in generation, necessitating additional denoising time. This challenge can be addressed by leveraging advancements in efficient denoising techniques and diffusion model architecture, potentially enhancing computational efficiency.

## 6 CONCLUSION

We proposed a complete system for stereoscopic video generation, using a video diffusion model and our *frame matrix* inpainting scheme. Given the fast adoption of video generation, our approach bridges the gap between the current ability to generate monocular and stereoscopic videos. In particular, we showed that our *frame matrix* formulation significantly advances the state-of-the-art for generative stereoscopic video, and can be adopted by existing and future video diffusion models.

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

## A APPENDIX

In the supplementary sections, we provide more studies and details of the proposed method.

- Sec. B provides a pseudocode describing our frame matrix denoising inpainting.

- Sec. C includes details on depth smoothing and warping-related artifacts handling.

- Sec. D provides more analysis of frame matrix.

- Sec. E elaborates details of human perception experiments and provides additional comparisons with Deep3D.

- Sec. F contains more studies of our method, including visualization of stereo effects, the ability to utilize temporal context, ablations on the number of cameras used, the efficacy of depth smoothing, intermediate results during the denoising process, consistency across different views, generalization ability of boundary re-injection, and comparisons with single-image view synthesis.

- Sec. G displays different trajectory results sampled from the frame matrix and cases in various scenarios.

## B ALGORITHM DETAILS

In the algorithm below, we present the detailed steps to denoise the Frame Matrix with spatial-temporal resampling, where we set $\mu_\theta(\mathbf{z}_t, c, t) = \frac{1}{\sqrt{1-\beta_t}}(\mathbf{z}_t - \frac{\beta_t}{\sqrt{1-\bar{\alpha}_t}}\epsilon_\theta(\mathbf{z}_t, c, t))$, following DDPM (Ho et al., 2020).

---
**Algorithm 1** Frame Matrix Inpainting
---
**Input:** $\mathbf{z}_T \sim \mathcal{N}(\mathbf{0}, \mathbf{I})$: Initial noisy latent maps
$\mathbf{z}_0$: Initial clean latent maps
**for** $t = T, ..., 1$ **do**
  **for** $n = 1, ..., N$ **do**
    **if** n is odd **then**
      Denoise time sequences $\{\mathbf{z}_{(:,v)t}|v = 0, ..., V\}$:
      **for** $v = 0, .., V$ **do**
        $\mathbf{z}^{\text{known}}_{(:,v)t-1} \sim \mathcal{N}(\sqrt{\bar{\alpha}_t}\mathbf{z}_{(:,v)0}, (1-\bar{\alpha}_t)\mathbf{I})$
        $\mathbf{z}^{\text{unknown}}_{(:,v)t-1} \sim \mathcal{N}(\mu_\theta(\mathbf{z}_{(:,v)t}, c, t), \Sigma_\theta(\mathbf{z}_{(:,v)t}, c, t))$
        $\mathbf{z}_{(:,v)t-1} = \mathbf{m}_{(:,v)} \odot \mathbf{z}^{\text{known}}_{(:,v)t-1} + (1 - m_{(:,v)}) \odot \mathbf{z}^{\text{unknown}}_{(:,v)t-1}$
      **end for**
    **else**
      Denoise view sequences $\{\mathbf{z}_{(s,:)t}|s = 0, ..., S\}$:
      **for** $s = 0, .., S$ **do**
        $\mathbf{z}^{\text{known}}_{(s,:)t-1} \sim \mathcal{N}(\sqrt{\bar{\alpha}_t}\mathbf{z}_{(s,:)0}, (1-\bar{\alpha}_t)\mathbf{I})$
        $\mathbf{z}^{\text{unknown}}_{(s,:)t-1} \sim \mathcal{N}(\mu_\theta(\mathbf{z}_{(s,:)t}, c, t), \Sigma_\theta(\mathbf{z}_{(s,:)t}, c, t))$
        $\mathbf{z}_{(s,:)t-1} = \mathbf{m}_{(s,:)} \odot \mathbf{z}^{\text{known}}_{(s,:)t-1} + (1 - m_{(s,:)}) \odot \mathbf{z}^{\text{unknown}}_{(s,:)t-1}$
      **end for**
    **end if**
    **if** $n < N$ **then**
      Add back one noise step for resampling:
      $\mathbf{z}_t \sim \mathcal{N}(\sqrt{1-\beta_{t-1}}\mathbf{z}_{t-1}, \beta_{t-1}\mathbf{I})$
    **end if**
  **end for**
**end for**
---

## C  DETAILS OF MONOCULAR VIDEO DEPTH WARPING

### C.1  DETAILS OF TEMPORAL DEPTH SMOOTHING

In practice, we use three consecutive frames to stabilize depth changes. At timestamp $t$, we first estimate optical flows $flow_{t-1 \to t}$ and $flow_{t+1 \to t}$, which help align the predicted depth maps at different timestamps. Finally, the refined depth at timestamp $t$ is obtained via convolving aligned depths $(D_{t-1 \to t}, D_t, D_{t+1 \to t})$ with a one-dimensional Gaussian kernel of shape $3 \times 1$.

### C.2  DETAILS OF HANDLING ISOLATED PIXELS AND CRACKS

**Multi-Plane projection.** Given RGB-D images, we warp them into a target camera view. Instead of projecting all pixels onto one image plane and handling occlusions using z-buffer, we divide the camera view space into multi-plane images $\{I_1^{step0}, ..., I_N^{step0}\}$ (N=4 in this paper) according to near and far depths, then each pixel is projected onto the image plane closest to it. We use $\{M_1^{step0}, ..., M_N^{step0}\}$ to indicate valid pixel positions on each image plane. By doing this, the foreground and background are separated in different planes, which makes artifacts (i.e., isolated points and entangled foreground and background in Fig. 6 left) easily to be handled.

**Remove isolated points.** Due to the inaccuracy of depth values around image boundaries, these pixels are warped into wrong positions leading to isolated pixels (see red box in Fig. 6 left). Intuitively, isolated pixels have no or very few neighbors, thus we detect isolated pixels based on this observation. Specifically, we apply convolution to each mask image $M_i^{step0}$ using a $3 \times 3$ kernel, after which isolated pixels are empirically determined where the values after convolution are less than 0.5. We remove these isolated pixels on both RGB and mask images to obtain updated $\{I_1^{step1}, ..., I_N^{step1}\}$ and $\{M_1^{step1}, ..., M_N^{step1}\}$.

**Remove cracks.** Since the depth image is not a watertight representation, the warped image usually contains cracks/holes that confuse foreground and background content. For example, the gray road can be seen through the dog's ear in Fig. 6 left. Similar to handling isolated pixels, we use a $3 \times 3$ Gaussian kernel to perform convolution on each mask image $M_i^{step1}$. When there are cracks, the values after convolution will be less than 1. In this paper, positions with no pixel values (0 in $M_i$) but with greater values than 0.2 after convolution are considered cracks. We fill cracks by interpolating nearby pixels in each image and obtain new multi-plane images $\{I_1^{step2}, ..., I_N^{step2}\}$ and $\{M_1^{step2}, ..., M_N^{step2}\}$.

After handling artifacts in each image plane, all image planes are blended into a final image (e.g., Fig. 6 ours left) in a back-to-front order using Eq. 11, where the content of front plane occludes content of the back plane.

$$I = I \times (1 - M_i^{step2}) + I_i^{step2} \times M_i^{step2}, \ for \ i \ in \ [N, ..., 1]. \tag{11}$$

## D  MORE ANALYSIS OF FRAME MATRIX

Based on the high-level motivation in the main paper, each frame $\mathbf{z}(i,j)$ in the frame matrix should be as consistent as the denoising results from both time and spatial directions. Mathematically:

$$L(i,j) = ||\mathbf{z}_{t-1}(i,j) - \mathbf{z}_{t-1}^{spatial}(i,j)||_2 + ||\mathbf{z}_{t-1}(i,j) - \mathbf{z}_{t-1}^{temporal}(i,j)||_2, \tag{12}$$

$$\mathbf{z}_{t-1}^{spatial} = \Theta^{spatial}(\mathbf{z}_t), \quad \mathbf{z}_{t-1}^{temporal} = \Theta^{temporal}(\mathbf{z}_t), \tag{13}$$

Where $\Theta$ is a pre-trained video diffusion model, $\Theta^{spatial}$ and $\Theta^{temporal}$ indicate denoising frame matrix in the time and spatial directions, respectively. Here, $L$ is a quadratic Least-Squares (LS) where the solution is as close as possible to all diffusion samples $\mathbf{z}_{t-1}^{spatial}(i,j)$ and $\mathbf{z}_{t-1}^{temporal}(i,j)$. In practice, we simultaneously denoise in spatial and time directions to obtain the optimal results.

# E    DETAILS OF HUMAN PERCEPTION STUDY

**Participants.** To evaluate the perceived quality of the generated stereoscopic videos, we recruited 20 participants (9 females) at least 18 years old ($\mu = 33, \sigma = 6.2$) with normal or corrected-to-normal vision at an anonymous institution via email lists and group communication software. The majority of participants had some experience with virtual reality. None of the participants was involved with this project prior to the user study.

**Study setup.** The study was conducted in a quiet meeting room with a commercial VR headset as the primary apparatus. The study software is implemented in Unity 2023.3.0b and we render stereoscopic videos with custom shaders on a $1.8m \times 1.0m$ quad that is three meters away from the participant in the world space, which occupies approximately 33.4 degrees in width and 18.92 degrees in height initially. Users have the freedom to move themselves within the meeting room to examine the stereoscopic video. This setup allowed participants to experience the stereoscopic videos in virtual reality settings and provided a controlled environment for the user study.

**Study protocol.** Each study session consists of a demographics interview with consent forms, a training session, and an evaluation session. To eliminate the ordering effect, we randomly counterbalanced all five methods for each video and assigned five random videos (out of 20 videos) with five conditions to each participant. However, since DynIBaR method failed to generate 13 videos, we collected a total of $5 \times 5 \times 20 - 13 \times 5 = 435$ evaluations from 20 participants, resulting in 100 human evaluations for each method except DynIBaR. During the training session, we randomly picked a video that was outside of the assigned videos to the participant and asked the participant to rate the stereoscopic effect, temporal consistency, graphical quality, and overall experience on a 7-point Likert scale (Likert, 1932), with 1 being the lowest, 7 being the highest, and 4 being the average. This procedure helps eliminate the novelty effect and calibrate the user's rating before the formal evaluation session. In the formal evaluation, we prompted the participant with the question like *"How would you like to rate the stereoscopic effect of the video on a 7-point scale, with 1 being the lowest, 7 being the highest, and 4 being the average?"* and asked the user the reason behind the rating.

**Metrics..** We evaluate the perceived quality of generated stereo videos based on three key aspects: 1. Stereo Effect. This refers to the perception of depth achieved by presenting slightly different images to each eye. A strong stereo effect makes objects appear closer or farther away, enhancing the 3D experience. Example questions: "How strong was the 3D effect in the video?" and "Which video felt more immersive due to the 3D effect?" 2. Temporal Consistency. This aspect assesses the smoothness of scene motion and the absence of artifacts such as jitter or ghosting over time. Example questions: "How smooth and natural did the motion of objects appear?" and "Did you notice any flickering, jumpiness, or distortions in the video?" 3. Graphical Quality. This evaluates the overall visual appeal of the video, including the quality of details, textures, lighting, and color fidelity. Example questions: "How would you rate the visual quality of the video?" and "Which video had more detailed and realistic textures?"

**Study results.** Overall, despite the missing data points for the DynIBaR method in some videos, Kruskal-Wallis tests (Kruskal & Wallis, 1952) reveals significant effects of group on all metrics respectively ($\chi^2 > 13.3, p < 0.01$): with stereo effect $\chi^2 = 186.3, p < 0.001$, temporal consistency $\chi^2 = 121.3, p < 0.001$, graphical quality $\chi^2 = 153.2, p < 0.001$, and overall experience $\chi^2 = 192.9, p < 0.001$. We further performed post-hoc tests using Mann-Whitney tests (Mann & Whitney, 1947) with Bonferroni correction, which revealed significant effects ($p < 0.05, |r| > 0.1$) for each pairwise comparison, except E2FGVI *vs.* ProPainters. Specifically, for Ours *vs.* E2FGVI, $p = 0.002$ on stereo effect, $p = 0.030$ on temporal consistency, $p < 0.001$ on graphical quality and overall experience. For Ours *vs.* ProPainter, $p = 0.004$ on stereo effect, $p = 0.017$ on temporal consistency, $p < 0.001$ on graphical quality and overall experience.

**Study findings.** Our results suggest that our methods achieve significantly better perceived stereoscopic effect than all other methods, while improvement in graphical quality and overall experience is more evident over stereoscopic effect; and stereo effect more evident over temporal consistency. During the study, we also observed many positive comments about our methods like "the contour is more clear", *"the graphics are sharper with fewer artifacts"*; however, we also observed negative or neutral feedback like *"some part really works and some parts don't: one side of the turtle face is wrong"*, and *"I see no difference (on the faces)"* from two participants. This suggests future research

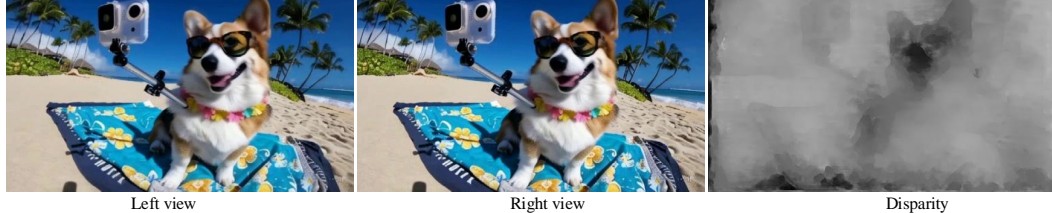

Left view        Right view        Disparity

Figure 8: **Results of Deep3D.** Deep3D does not provide the function to change the stereo baseline, and the vague disparity map on the right side demonstrates its weak stereo effects.

to investigate holistic perceptual consistency in stereoscopic videos and finetune models for special subjects like human beings.

**Additional user study on Ours *vs.* Deep3D.** We further conducted a human evaluation between Ours and Deep3D across the same metrics with a total of 190 random evaluations over 20 random videos, following the same protocol. Pairwise Mann-Whitney tests with Bonferroni correction reveal significant effects on stereo effect ($p < 0.001$), overall experience ($p < 0.001$), and temporal consistency ($p = 0.015$). We found our method outperforms Deep3D in stereo effect and overall experience, yet falling slightly short in temporal consistency.

Similar to Fig. 10, we visualize Deep3D's disparity map in Fig. 8. The vague disparity map in the third column demonstrates weak stereo effects, which matches the statistic results in Table 3. By manually modifying the disparity map or changing the stereo baseline, 3D effects may become apparent. However, Deep3D does not support these functions.

|  | Deep3D | Ours | $p$-value | $|r|$ (effect size) |
|---|---|---|---|---|
| Stereo Effect ↑ | 2.29 (1.63) | **5.29** (1.09) | $< 0.001$ *** | 0.60 (large) |
| Temporal Consistency ↑ | **5.37** (1.23) | 5.06 (1.25) | 0.015 * | 0.49 (medium) |
| Image Quality ↑ | **5.27** (1.17) | 5.12 (1.19) | 0.103 | 0.10 (small) |
| Overall Experience ↑ | 3.68 (1.36) | **5.08** (1.09) | $< 0.001$ *** | 0.57 (large) |

Table 3: **Human perception.** This table reports results of human perception experiments as mean (std) between Deep3D and Ours. Our method outperforms Deeph3D in stereo effect and overall experience, yet falls slightly short in temporal consistency. Mann-Whitney tests with Bonferroni correction reveals significant effects on stereo effect ($p < 0.001$, $Z = -8.24$), overall experience ($p < 0.001$, $Z = -7.92$), and temporal consistency ($p = 0.015$, $Z = -6.72$).

## F  MORE EXPERIMENTS

**Stereo effects visualization.** To further verify the quality of stereo effects, we employed the Stereo Transformer (Li et al., 2021a) to predict disparity maps for our generated stereo videos. As illustrated in Fig. 9 (a), the resulting disparity maps are sharp, with a clear distinction between foreground and background elements, indicating the reasonable 3D effect of our results. Additionally, since a majority of pixels are generated by warping the monocular estimated depth and the diffusion model tends to preserve the known pixels, the generated binocular pair maintains the monocular depth very well. This is verified by the example in Fig. 9 (b), which shows disparity maps from monocular estimation (first column) and binocular estimation (second column), and the difference between them (last column). The quantitative difference is 0.63 pixels, which indicates overall good consistency.

**Ability to utilize temporal context for inpainting.** Our method is able to harmonize image contents between different temporal frames during inpainting and thus enhance temporal consistency. Figure 11 shows one example. When inpainting the right-view frame at $t$, our method successfully creates content that is consistent with the left-view frame at $t + 1$ (see the generated character "R" in the disoccluded region). Note that such consistency is maintained automatically thanks to frame matrix based denoising, since all temporal frames are taken into account.

**Number of cameras used.** We reduce the number of cameras between left and right views and show results in Fig. 12. Artifacts (e.g., the fifth leg) tend to occur when the number of cameras is fewer

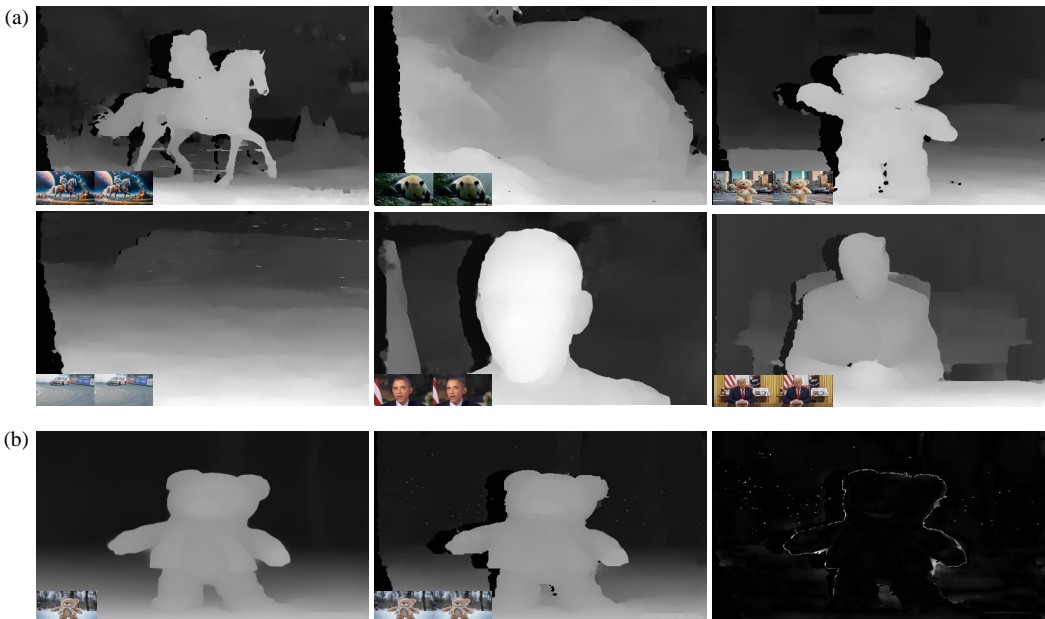

Figure 9: **Stereo effects visualization.** (a) The generated stereo results are used for predicting disparity maps (Li et al., 2021a). The distinction between foreground and background elements indicates reasonable 3D effects. (b) Disparity maps are estimated from monocular and binocular images. The error map on the right side demonstrates their consistency.

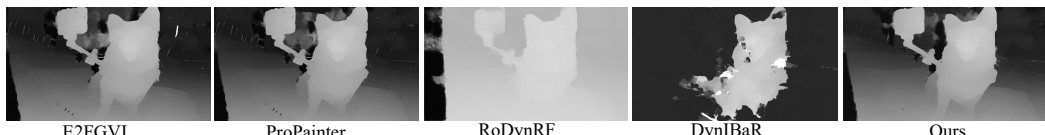

Figure 10: **Estimated disparity maps of different methods.** The disparity values change reasonably in our method.

than four in this horse case, as a limited number of cameras can lead to rapid changes between video frames that exceed the capabilities of current video generation models.

**Efficacy of temporal depth smoothing.** In Fig. 14, we track a pixel across video frames and display their depth changes with and without using our depth smoothing operation. From Fig. 14 right side, our approach effectively stabilizes depth changes across frames.

**Results at different denoising steps.** In Fig. 13, we present results at different denoising steps to enhance the understanding of the denoising inpainting process. Without the disocclusion boundary re-injection, noticeable artifacts emerge throughout the denoising process. In contrast, our method progressively fills disoccluded regions with harmonious content.

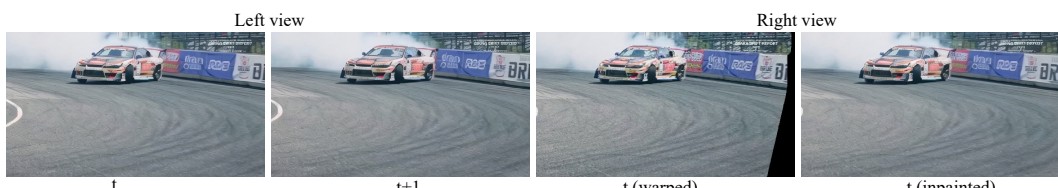

Figure 11: **Ability to utilize unobserved content.** Left view: two consecutive images observed by the left view. Right view: the warped and inpainted images at time t. Note that the black region is inpainted with the character "R", matching the characters in the second image at time t+1.

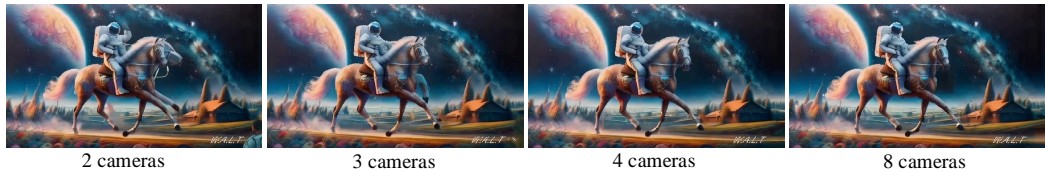

| 2 cameras | 3 cameras | 4 cameras | 8 cameras |

Figure 12: **Number of cameras between left and right views.** Artifacts arise when the number of cameras is too small to exceed the capabilities of video generation models.

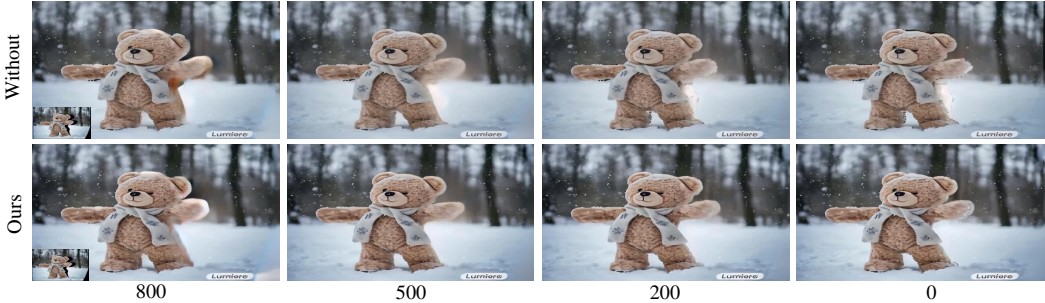

| 800 | 500 | 200 | 0 |

Figure 13: **Results at different denoising steps.** Without the use of disocclusion boundary re-injection, artifacts persist throughout the denoising process. In contrast, our method gradually fills disoccluded regions with harmonious content.

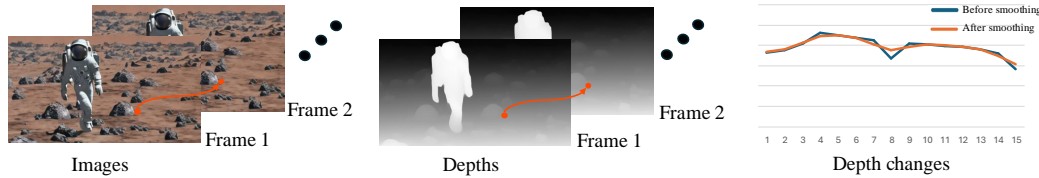

Figure 14: **Temporal depth smoothing.** We track a pixel's depth values across frames and visualize the depth changes. By applying smoothing operations, the depth changes more smoothly.

**Consistency across different viewpoints.** In Fig. 17, the first row is warped images under different camera views. We generate each view independently and show results in the second row, where the content is not consistent across different views, such as the dragon's wing. With the help of the frame matrix, which also regularizes generation in the direction of camera motion, our results in the third row are more consistent.

**Apply boundary re-injection to image inpainting (Avrahami et al., 2023).** Our method is also applicable to previous image inpainting approaches that utilize latent diffusion backbones. In Fig. 19, the masked region is inpainted as a stone, and our method showcases a smoother transition between original and new content (e.g., the grassland), thanks to the boundary re-injection designs that update latent features around the mask boundary.

**Compare with single-image view synthesis methods.** Stereoscopic video can be obtained by dealing with each frame individually using a single-image view synthesis method. Here, we compare our approach with SVM (Tucker & Snavely, 2020) and AdaMPI (Han et al., 2022), and show results at different time stamps in Fig. 20. At a specific time stamp (e.g., $t_1$), both SVG and AdaMPI produce blurry content next to the character "B", whereas our method generates a sharp character "R". This improvement is attributable to our approach's ability to leverage information from other frames (i.e., left-view frame at time $t_2$). Moreover, the results from AdaMPI and SVM are not temporally consistent because each frame is processed individually. For example, the content next to the character "B" differs between $t_1$ and $t_2$. On the contrary, our approach leverages the video generation model to ensure temporal consistency, allowing the characters in our results to align with those in the left-view frames. More cases are included in Fig. 22. Quantitative comparisons are displayed in Table 4.

|          | SVM   | AdaMPI | Ours      |
|----------|-------|--------|-----------|
| DOVER ↑  | 0.215 | 0.245  | **0.584** |
| FVD ↓    | 784   | 718    | **599**   |

Table 4: **Compare with single-image view synthesis.** Our method outperforms SVM and AdaMPI on aesthetic scores, measured by DOVER and FVD.

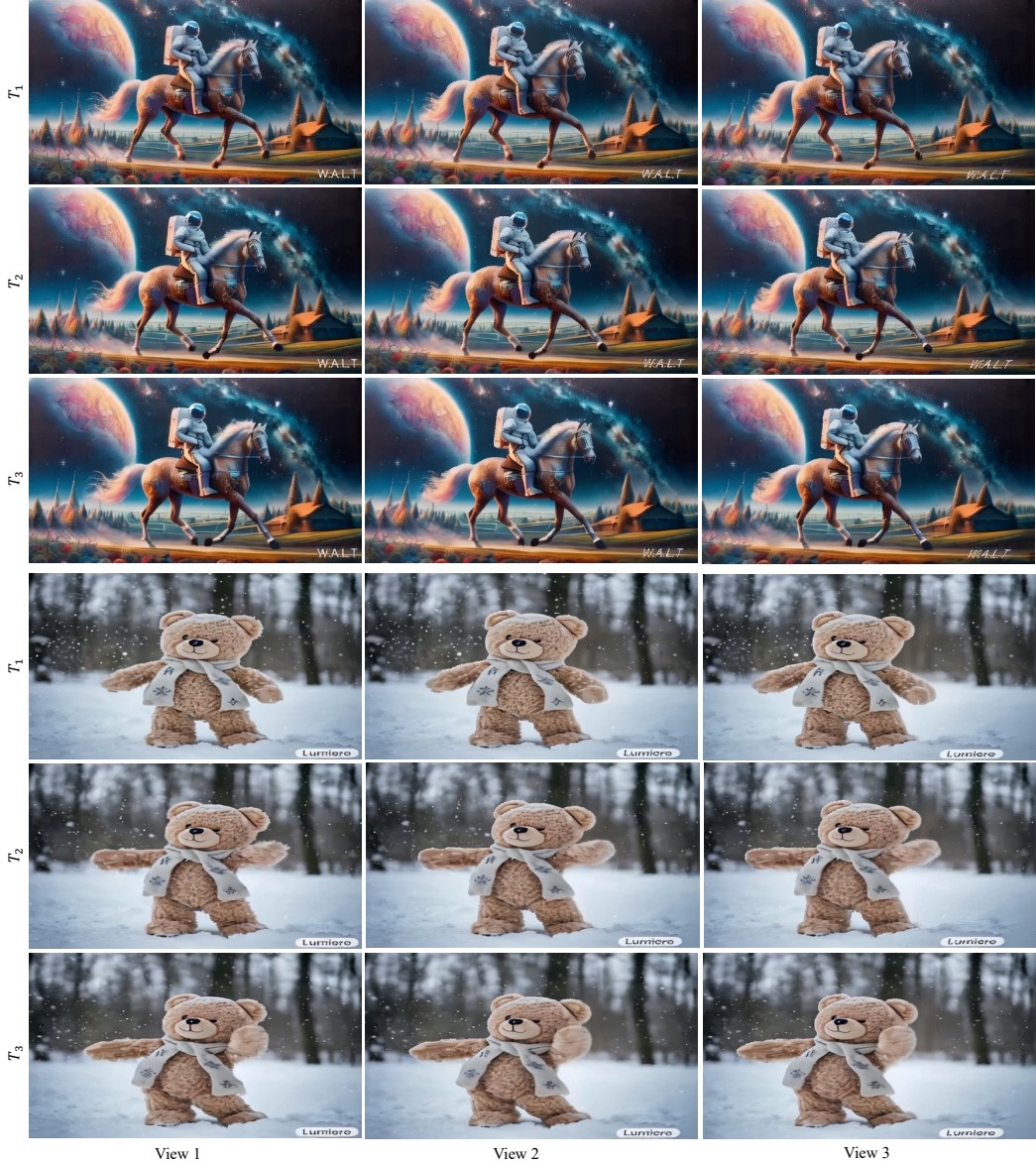

Figure 15: **Frames sampled from frame matrix.** In both cases, each column represents a generated video from a camera, while each row corresponds to the generated frames from different cameras at a specific timestamp.

## G   MORE RESULTS

**Other trajectories in frame matrix.** In the main paper, we show generated left and right views. Here, we additionally show the results of other trajectories. In Fig. 15, we selectively display frames generated within the frame matrix at different timestamps (3 out of 16) in different camera views (3

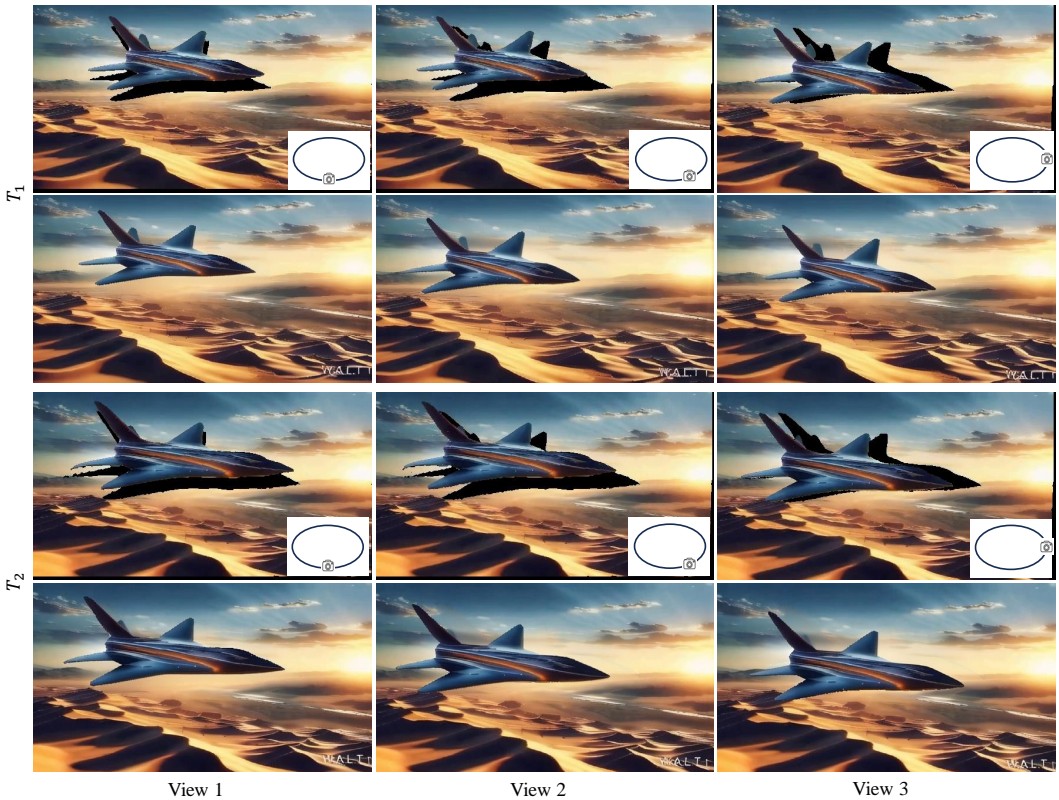

Figure 16: **Frames sampled from frame matrix constructed using a spiral trajectory.** Warped and generated frames in different cameras at different timestamps.

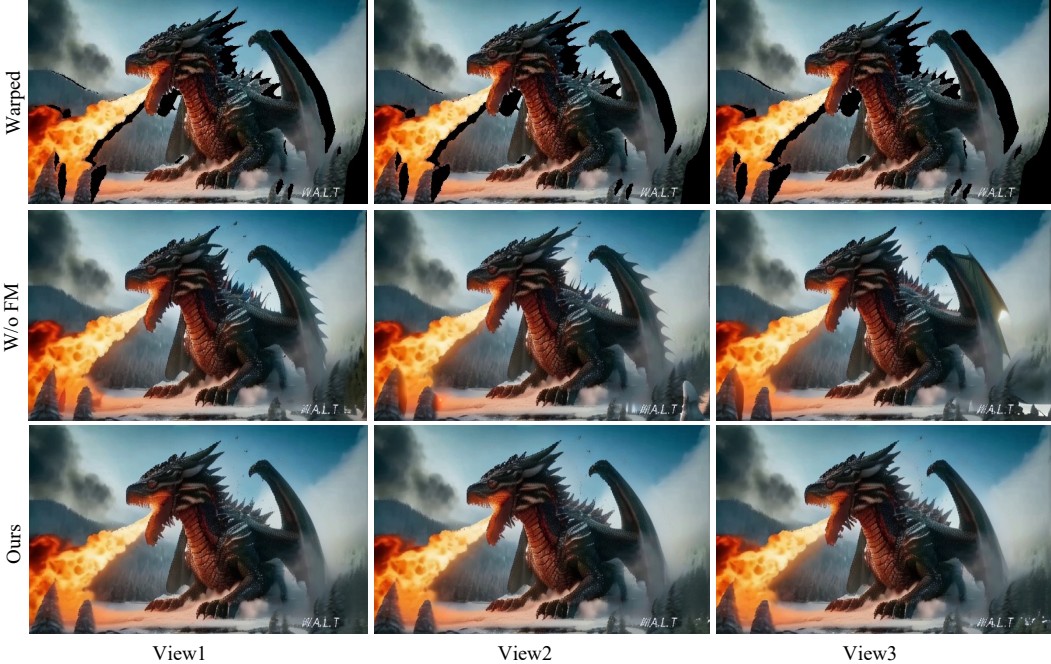

Figure 17: **Frame matrix benefits consistent content generation.** When each view is processed independently, the dragon's wing varies across different camera viewpoints.

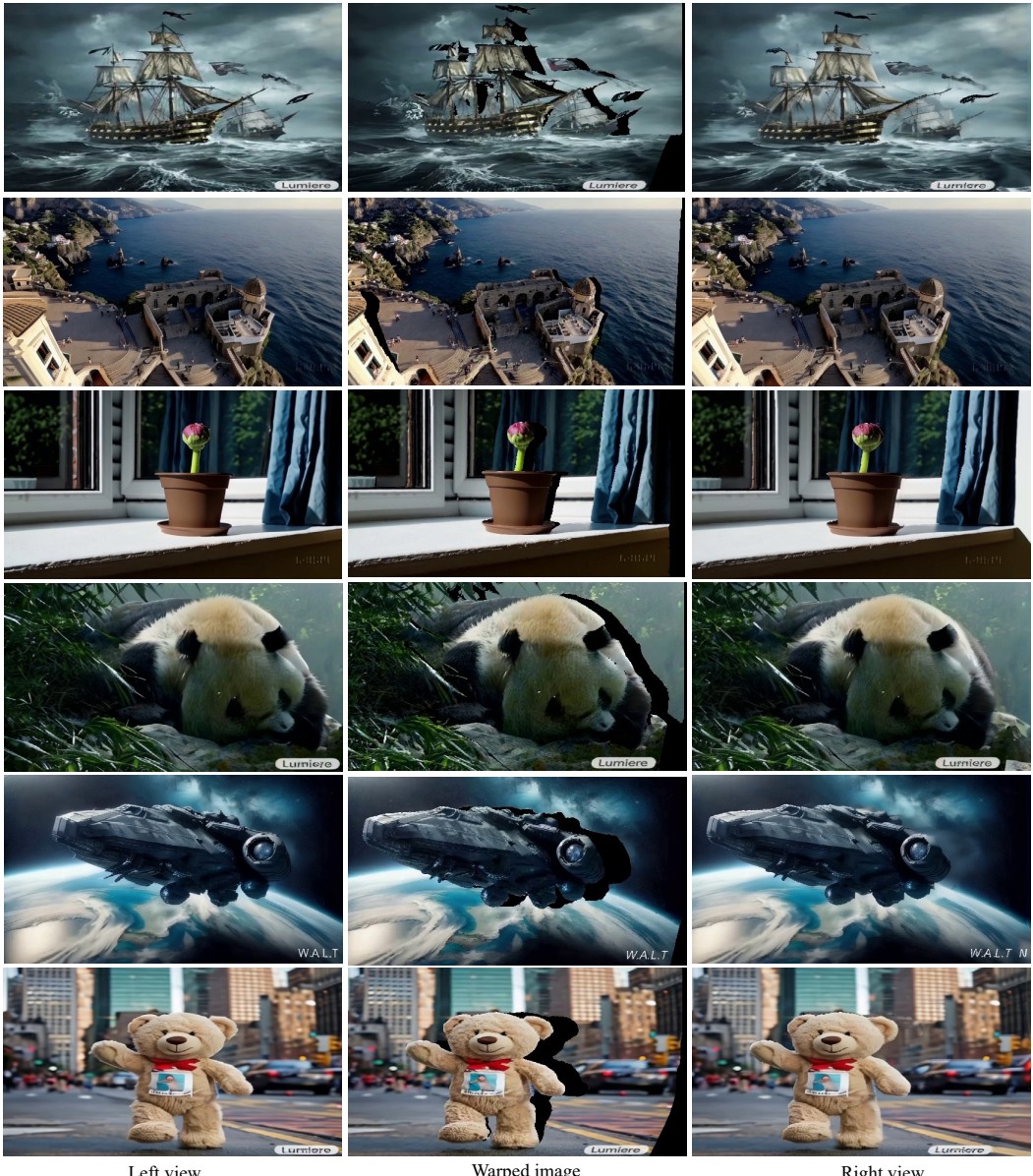

Left view                  Warped image                  Right view

Figure 18: **We display more generated results in various scenarios.**

out of 8). From the results, both foreground and background content are coherent across different frames. Furthermore, instead of building the frame matrix using a camera moving from left to right, we move the camera along a spiral trajectory. In Fig. 16 first and third rows, we selectively show the warped images in different camera views (3 out of 16), where disocclusions appear around the plane. Under each warped image, we display the corresponding image with disocclusions filled.

**Diverse outputs.** As shown in Fig. 21, the output is not unique because the disoccluded regions can be generated with different plausible contents, such as cliffs or trees.

**More cases.** In this part, more generated results are displayed in Fig. 18. The proposed method works on different scenarios, such as the beautiful church, imaginary scenes, and ships in the storm where the whole scene is dynamic. The high-quality generated results in Fig. 18 right column demonstrate the generalization ability of the proposed method. Moreover, Fig. 23 includes results from two complex real-world scenes. The dolphin case features multiple moving objects, while the train case

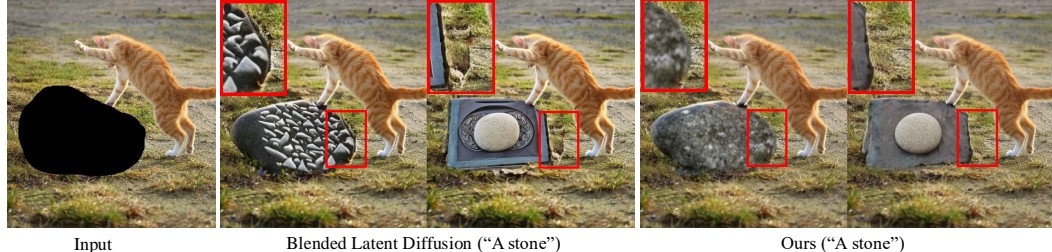

Figure 19: **Boundary re-injection in image inpainting.** Our approach (blended latent diffusion + disocclusion boundary reinjection) provides a smoother transition between original and inpainted content, such as the grassland.

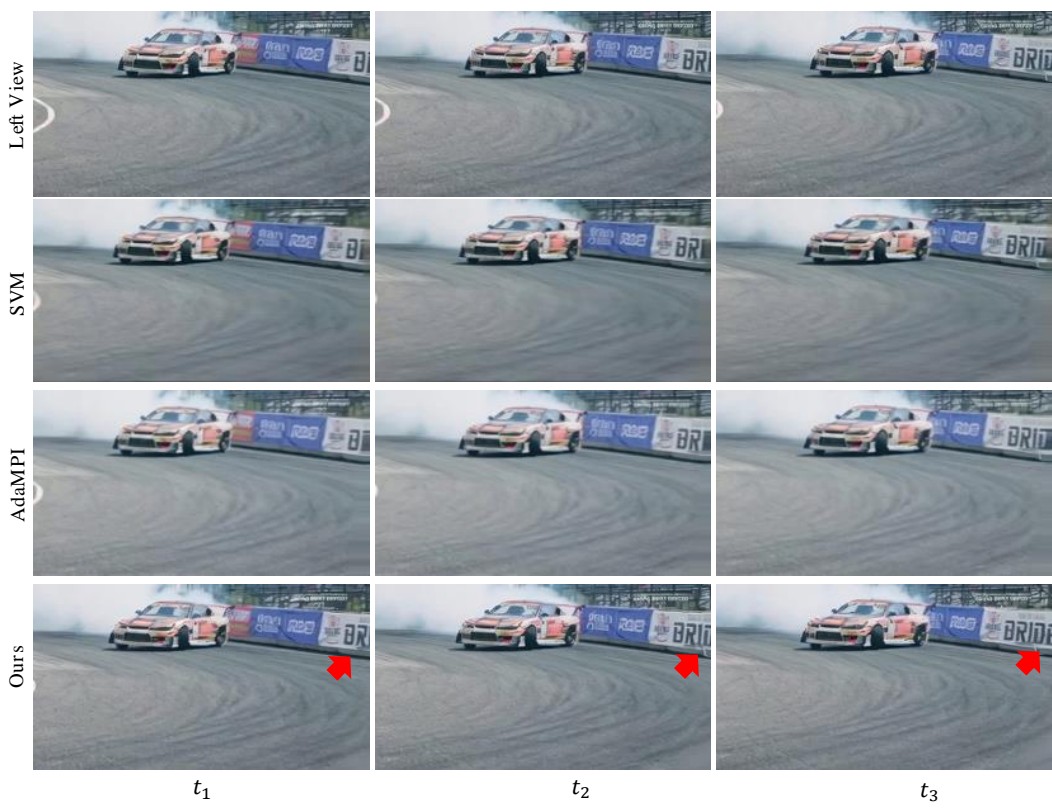

Figure 20: **Compare with single-image view synthesis.** SVM and AdaMPI tend to produce blurry and temporally inconsistent results, please note the content next to the character "B" across different time stamps.

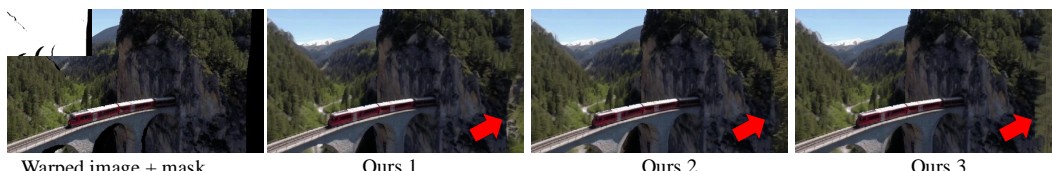

Figure 21: **Diverse outputs.** The disoccluded regions can be generated with different content.

involves a fast-moving object. Our method performs effectively in both scenarios and outperforms results without adopting boundary re-injection design.

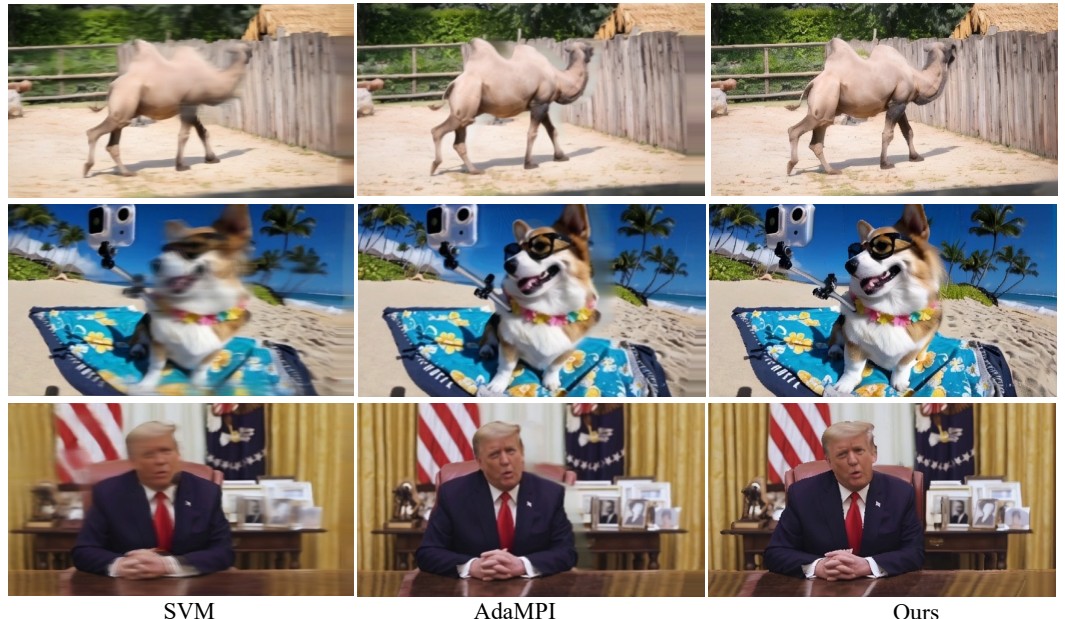

|  |  |  |
|:---:|:---:|:---:|
| SVM | AdaMPI | Ours |

Figure 22: **More comparisons with single-image view synthesis.** SVM and AdaMPI tend to produce blurry results in disoccluded regions.

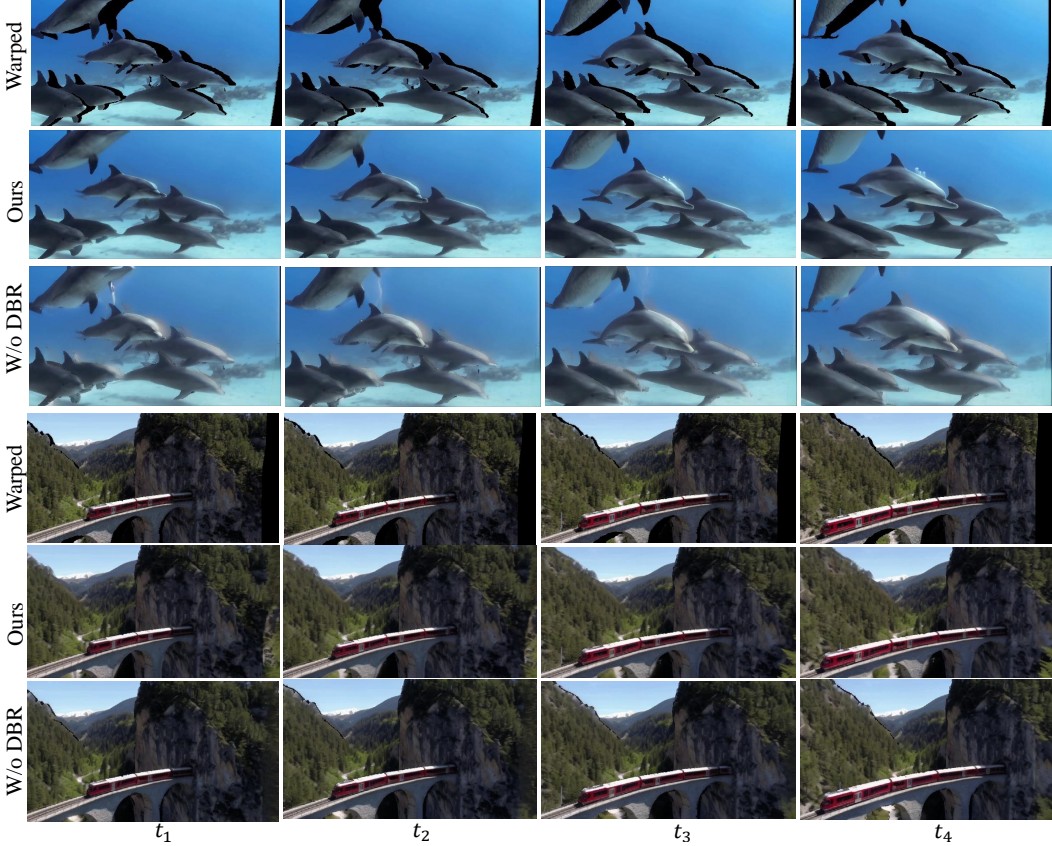

Figure 23: **Complex real-world scenes.** Our method is effective in real-world scenes featuring multiple moving objects, such as dolphins, as well as fast-moving objects, like trains. Without using boundary re-injection, results have obvious artifacts.

