# OpenReview forum: "SVG: 3D Stereoscopic Video Generation via Denoising Frame Matrix"
_ICLR.cc/2025/Conference — ICLR 2025 Poster_

### Official Review · Reviewer_dHEi · 2024-10-16

**Soundness:** 3
**Presentation:** 1
**Contribution:** 2
**Rating:** 6
**Confidence:** 5

**Summary:**

This paper proposes a training-free framework to generate stereo video from single video sequence by using retrained video generation model. The proposed frame matrix leverages the power of video generation model and the joint optimization idea to generate semantically consistent and temporally smooth content, which offers valuable insights and can provide reference directions for future work.. However, besides the frame matrix, the other modules mentioned in methods are more like tricks and are all followed by existing works. The extensive results show that this paper achieve SOTA performance in training-free manner. However, the drawbacks are obvious, the performance is significantly worse than the training methods, and the process is slow due to the need for multiple iterations.

**Strengths:**

1. This paper proposes a training-free manner to generate stereo video from monocular video and achieve SOTA performance in training-free manner.

2. The proposed denoising frame matrix uses pre-trained video generation model as the inpainting model, which is the first one to do this in stereo video generation field and offers insight for leveraging video generation model to assist this task.

**Weaknesses:**

1. The analysis for denoising frame matrix is insufficient. Although the authors provide the reason for using video generation model, the theory and the high-level reason of why it works are not clear, please show the theoretical analysis of the proposed frame matrix.

2. Lack citations for the methodology followed by other methods. The presentation in the paper contains certain misleading and deceptive elements. Line 153-161, the viewpoint transfer part is widely used in novel view synthesis [R1, R2] and stereo image generation [R3]. Additionally, the proposed boundary re-injection is just a trick, which can not be a contribution. These parts should be revised for better presentation.

[R1] Tucker, R.; and Snavely, N. 2020. Single-view view synthesis with multiplane images. In Proceedings of the IEEE/CVF Conference on Computer Vision and Pattern Recognition, 551–560.

[R2] Han, Y.; Wang, R.; and Yang, J. 2022. Single-view view synthesis in the wild with learned adaptive multiplane images. In ACM SIGGRAPH 2022 Conference Proceedings, 1–8.

[R3] Wang, X.; Wu, C.; Yin, S.; Ni, M.; Wang, J.; Li, L.; Yang, Z.; Yang, F.; Wang, L.; Liu, Z.; et al. 2023b. Learning 3D Photography Videos via Self-supervised Diffusion on Single Images. arXiv preprint arXiv:2302.10781.

3. Insufficient evaluation metics. In [R1], [R2], [R3], they will use SSIM, PSNR, LPIPS evaluation metrics in novel-view image, which can also be used in stereo video generation. What's more, the temporal consistency is just justified in user study, the quantitative results such as Fréchet Video Distance should also be given.

4. Lack comparison with highly related methods. The methods compared in this paper is too simple. As mentioned in Q2, [R1], [R2], [R3] are the highly related stereo image generation methods. Although they need to train, they own the advantages of high performance and fast inference speed, the authors should give the result in the paper and clarify the strength of their methods.

**Questions:**

See the weakness.

**Details Of Ethics Concerns:**

None.

---

> ### Author Response · Authors · 2024-11-23
>
> **Q1. Claim boundary re-injection as a contribution.**
>
> 1) To the best of our knowledge, we are the first to report that unknown regions can negatively impact the inpainting process when using latent diffusion models. Previous works like blended latent diffusion[1] and Repaint[2], which assume the input image to be complete (no unknown regions) or use pixel diffusion models for image inpainting, thus do not face the boundary issues reported in our paper. However, the input image is not always complete (e.g., warped images with unknow regions) and latent diffusion is widely used in generation. To address latent-diffusion-based inpainting, we propose the disocclusion boundary re-injection design that effectively and significantly improves the quality of inpainting results.
>
> 2) Additionally, we demonstrate that boundary re-injection can be incorporated into existing latent diffusion-based inpainting methods, such as blended latent diffusion, to further enhance their performance in inpainting unkonwn regions. The effectiveness of this enhancement is illustrated in Fig. 18.
>
> [1] Omri Avrahami, Ohad Fried, and Dani Lischinski. Blended latent diffusion. ACM Transactions on
> Graphics (TOG), 42(4):1–11, 2023.
>
> [2] Andreas Lugmayr, Martin Danelljan, Andres Romero, Fisher Yu, Radu Timofte, and Luc Van Gool.
> Repaint: Inpainting using denoising diffusion probabilistic models. In Proceedings of the
> IEEE/CVF conference on computer vision and pattern recognition, pp. 11461–11471, 2022.
>
> **Q2. Contributions beyond the frame matrix**
>
> The frame matrix is undoubtedly the most significant contribution of this work; however, the pipeline also depends on other key modules to robustly generate high-quality, temporally consistent stereo videos with strong 3D perception. By incorporating boundary re-injection, we enhance the visual quality and smoothness of the disocclusion boundaries. Other modules (not emphasized as contributions), including depth smoothing and removal of isolated points and cracks, are crucial for reducing depth perturbations across frames and addressing problematic artifacts caused by perspective projection.
>
> **Q3. The process is slow due to the need for multiple iterations.**
>
> Yes, efficiency is not our primary focus. We have discussed this limitation and potential solutions in the limitations section.
>
> **Q4. The performance is significantly worse than the training methods, and comparisons with suggested training methods.**
>
> 1) In this paper, our method outperforms previous video inpainting approaches, including ProPainter [3] and E2FGVI [4], as well as the mono-to-stereo conversion method Deep3D [5], all of which are training-based methods. Could you please provide more details regarding the statement, "The performance is significantly worse than the training methods"? We would like to clarify any potential misunderstandings.
>
> 2) Additionally, we added comparisons between our method and the suggested training methods (i.e., AdaMPI [R2] and SVM [R1]) in the rebuttal, with results presented in Fig. 20 and Fig. 21. These methods, designed for single-image novel view synthesis, process frames captured at different timestamps independently. As a result, they produce temporally inconsistent outputs and cannot leverage information across frames at varying timestamps. Since the code for [R3] has not been released, we are unable to provide a direct comparison. However, this method faces similar challenges, as [R3] is also designed for stereo image generation.
>
> 3) We show quantitative comparisons in the following table. Our method outperforms SVG and AdaMPI.
>
>     |Metric|SVM|AdaMPI|Ours|
>     | :-----:| :----: | :----: | :----: |
>     |DOVER [6] ↑|0.215|0.245|0.584|
>     |FVD [7] ↓|784|718|599|
>
> [3] Shangchen Zhou, etc. Propainter: Improving
> propagation and transformer for video inpainting. In Proceedings of the IEEE/CVF International
> Conference on Computer Vision, pp. 10477–10486, 2023
>
> [4] Zhen Li, etc. Towards an end-to-end
> framework for flow-guided video inpainting. In Proceedings of the IEEE/CVF conference on
> computer vision and pattern recognition, pp. 17562–17571, 2022b
>
> [5] Junyuan Xie, etc. Deep3d: Fully automatic 2d-to-3d video conversion
> with deep convolutional neural networks. In Computer Vision–ECCV 2016: 14th European
> Conference, Amsterdam, The Netherlands, October 11–14, 2016, Proceedings, Part IV 14, pp.
> 842–857. Springer, 2016.
>
> [6] Haoning Wu, etc. Exploring video quality assessment on user generated contents from
> aesthetic and technical perspectives. In Proceedings of the IEEE/CVF International Conference on
> Computer Vision, pp. 20144–20154, 2023a.
>
> [7] Thomas Unterthiner, etc. Fvd: A new metric for video generation. 2019.

---

> > ### Author Response · Authors · 2024-11-23
> >
> > [R1] Tucker, R.; and Snavely, N. 2020. Single-view view synthesis with multiplane images. In Proceedings of the IEEE/CVF Conference on Computer Vision and Pattern Recognition, 551–560.
> >
> > [R2] Han, Y.; Wang, R.; and Yang, J. 2022. Single-view view synthesis in the wild with learned adaptive multiplane images. In ACM SIGGRAPH 2022 Conference Proceedings, 1–8.
> >
> >  [R3] Wang, X.; Wu, C.; Yin, S.; Ni, M.; Wang, J.; Li, L.; Yang, Z.; Yang, F.; Wang, L.; Liu, Z.; et al. 2023b. Learning 3D Photography Videos via Self-supervised Diffusion on Single Images. arXiv preprint arXiv:2302.10781
> >
> >
> > **Q5. High-level reason and more analysis of frame matrix.**
> >
> > 1) High-level motivations. In practice, 3D stereoscopic videos can be produced by recording with two cameras
> > (time-direction videos). Since both cameras are capturing the same scene, gradually moving the left camera toward the right camera also results in a coherent video (spatial-direction video). Likewise, it is essential to consider both the time and spatial directions when generating the 3D stereoscopic video to ensure that both perspectives represent the same scene. For this purpose, we propose the frame matrix, allowing the denoising process to be performed simultaneously along both spatial and temporal dimensions using a pre-trained video diffusion model, which has learned spatial and temporal priors from diverse static and dynamic scenes.  Additionally, compared to direct inpainting a large region all at once, gradually expanding the inpainting area (spatial direction) tends to be easier and can lead to more stable and plausible results.
> >
> > 2) Theoretical analysis. Based on the above high-level motivation, each frame in the frame matrix should be as consistent as the denoising results from both time and spatial directions. Mathematically:
> >
> >      $$
> >       L(i, j) = ||z_{t-1}(i, j) - z_{t-1}^{spatial}(i, j)||^2 + ||z_{t-1}(i, j) - z_{t-1}^{time}(i, j)||^2,
> >      $$
> >
> >      $$
> >      z_{t-1}^{spatial} = \Theta^{spatial}(z_t) ,
> >      $$
> >
> >      $$
> >      z_{t-1}^{time} = \Theta^{time}(z_t) ,
> >      $$
> >
> > Where $\Theta$ is a pre-trained video diffusion model, $\Theta^{time}$ and $\Theta^{spatial}$ indicate denoising frame matrix $z$ in the time and spatial directions respectively. Here, $L$ is a quadratic Least-Squares (LS) where the solution is as close as possible to all diffusion samples $z_{t-1}^{time}(i, j)$ and  $z_{t-1}^{spatial}(i, j)$. In practice, we simultaneously denoise in spatial and time directions to obtain the optimal results.
> >
> > **Q6. Lack citations**
> >
> > We have included the missing articles in our new version (lines 689-691, 576-578, 700-701).
> >
> > **Q7. The presentation contains certain misleading and deceptive elements.**
> >
> > Could you please provide more details? We'd like to clarify any misunderstandings.
> >
> > **Q8. Line 153-161, the viewpoint transfer part is widely used in novel view synthesis [R1, R2] and stereo image generation [R3].**
> >
> > Lines 153-161 describe the overall workflow of our pipeline for creating stereoscopic videos, where we adopt the warp-then-inpaint approach, for which we do not claim contributions. Our primary focus and contribution lie in addressing the challenges of filling disoccluded regions with semantically reasonable and high-quality content by leveraging pre-trained video generation models in a zero-shot manner. To achieve this, we propose a novel frame matrix representation and a re-injection scheme.
> >
> > **Q9. Insufficient evaluation metrics.  In [R1], [R2], [R3], they will use SSIM, PSNR, LPIPS evaluation metrics in novel-view image, which can also be used in stereo video generation., the quantitative results such as Fréchet Video Distance should also be given.**
> >
> > 1) As shown in Fig. 22, the generated contents in disoccluded regions are not unique but visually reasonable. Therefore, metrics such as PSNR, SSIM, and LPIPS, which are calculated for measuring reconstruction error instead of generation quality, are not suitable for this task.
> >
> > 2) We added the suggested FVD in Table 2.
> >
> > **Q10. Lack of comparison with highly related methods [R1, R2, R3]. The authors should give the results in the paper and clarify the strength of their methods.**
> >
> > 1) Please refer to Q4 for comparisons.
> >
> > 2) Methods [R1, R2, R3] are not designed for stereoscopic video and may encounter temporal consistency issues, as they process a single image and thus cannot leverage temporal information across frames (see Fig. 20). In contrast, we leverage a video diffusion model that enables effective utilization of temporal information.
> >
> > 3) Collecting stereoscopic videos for training these methods is costly, and insufficient training data can lead to performance issues due to the domain gap. In contrast, the pre-trained video diffusion model, trained on a vast amount of data, demonstrates great generalizability.

---

> > > ### Comment · Reviewer_dHEi · 2024-11-24
> > >
> > > Thank you for the response, it mainly addresses my concerns but only one small suggestion.
> > >
> > > For Q8, the author should add the citation in LIne 156-157 ``according to the stereoscopic baseline''. Otherwise, It feels like a sense of concealing the contributions of others.

---

> > > > ### Author Response · Authors · 2024-11-24
> > > >
> > > > Thank you for providing us with more details and suggestions. We added [R2] and [R2] as references. Please check our updated PDF.

---

> > > > > ### Author Response · Authors · 2024-11-24
> > > > >
> > > > > [R2] and *[R3] as references

---

> > ### Comment · Reviewer_dHEi · 2024-11-24
> >
> > Thank you for the response. It addresses my concern about the experiment's integrity, but not for the contribution of boundary re-injection and the limitations of this paper.
> >
> > For boundary re-injection, I understand the implementation and the effectiveness. However, this module is essentially just an engineering implementation, regardless of whether others have implemented it. It lacks novelty and is insufficient to serve as a contribution for an academic paper. Of course, I fully acknowledge the innovation of the frame matrix. I suggest the author focus more on the analysis and discussion of the frame matrix and minimize the emphasis on boundary re-injection as much as possible.
> >
> > For the inference speed limitation. It's not enough for the author to simply mention this issue in the limitations section and then disregard it. In any scenario, excessively long inference times for 3D video generation are unacceptable, and it might be more effective to collect data to address this problem. Of course, I fully acknowledge the contributions of this paper and understand that development takes time. However, from my perspective, the time overhead caused by this optimization approach is an almost inevitable issue with little room for further optimization. I hope the author can face this problem and offer some potential solutions.
> >
> > I will raise my point as the experiment result is great and address my concerns, but addressing the suggestion above will elevate the value of this paper to a new level.

---

> > > ### Author Response · Authors · 2024-11-25
> > >
> > > Thanks for your suggestions.
> > >
> > > - We will moderately reduce the discussion on boundary re-injection to allocate more space for analyzing the frame matrix in the final version.
> > >
> > > - Except for reducing the denoising steps to speed up inference, which is a hot topic in diffusion models, using fewer cameras between left and right views (suggested by reviewer P3fm) is also viable. As shown in Fig. 19, employing 4 cameras, which halves the computation efforts, can produce competitive results.
> > >
> > > - The proposed frame matrix is ​​a representation that connects the 4D scene with the 2D generative models. Currently, the frame matrix is ​​used to generate reasonable stereoscopic videos, but can be further explored, such as multi-view video generation and zero-shot consistent 3D/4D content editing, which we consider as our future work. Furthermore, exploring the use of generated data for training [1] is worthwhile; we could potentially utilize our generated stereoscopic videos to train a diffusion model, thereby addressing the scarcity of 3D and multi-view videos.
> > >
> > > [1] He, etc. Is synthetic data from generative models ready for image recognition? ICLR 2023.

---

### Official Review · Reviewer_zJCL · 2024-11-02

**Soundness:** 3
**Presentation:** 3
**Contribution:** 3
**Rating:** 8
**Confidence:** 3

**Summary:**

The paper addresses a gap in the generation of stereoscopic videos, particularly corresponding to the advancements in VR/AR technologies. The authors introduce an interesting pose-free and training-free framework that aims to improve the generation of high-quality 3D stereoscopic videos from monocular video inputs. The author utilizes video generation models to enhance 3D consistency while addressing challenges such as occlusion and temporal stability.

**Strengths:**

- The paper is well-written and easy to follow.
- The proposed frame matrix representation is reasonable, and the extensive experiments support its functionality. Therefore, this paper is sufficiently novel.
- The proposed method demonstrates a strong understanding of the challenges specific to 3D video generation, including issues with depth estimation and video inpainting.
- I believe that sufficient experiments and ablation studies are presented to support the approach.

**Weaknesses:**

The main weaknesses of the proposed method are the disocclusion boundary artifacts, slightly lower temporal consistency compared to Deep3D, and the need for further improvements in holistic perceptual consistency, especially for certain subjects like human faces.

**Questions:**

Missing some reference about multi-view synthesis, please consider reference them:

[1] Chen, Zilong, et al. "V3d: Video diffusion models are effective 3d generators." arXiv preprint arXiv:2403.06738 (2024).
[2] Zuo, Qi, et al. "Videomv: Consistent multi-view generation based on large video generative model." arXiv preprint arXiv:2403.12010 (2024).

---

> ### Author Response · Authors · 2024-11-23
>
> **Q1. The disocclusion boundary artifacts.**
>
> We agree that disocclusion boundary artifact is a major challenge in obtaining high-quality results. In Fig. 5 and Fig. 18, our proposed boundary reinjection design can mitigate artifacts around disocclusion boundary. In the future, more advanced video generation models might be beneficial for better recognizing and addressing these artifacts.
>
>
> **Q2. Slightly lower temporal consistency compared to Deep3D.**
>
> In our experiment, Deep3D tends to produce a right-view video that closely resembles the provided left-view video. However, while more consistent results can be achieved, this comes at the cost of significantly sacrificing their 3D effects, as visualized in Fig. 9.
>
>
> **Q3. The need for further improvements in holistic perceptual consistency, especially for certain subjects like human faces.**
>
> We agree that holistic consistency is a crucial factor for immersive 3D, especially for the human face that is fragile to artifacts. To address this issue, employing more advanced video generation and depth estimation models that yield consistent results is a viable approach. In addition to changing models, incorporating explicit 3D reconstruction techniques, as referenced in the papers you mentioned, could enhance the holistic consistency of the objects. We plan to explore this topic further and feature it in our future work.
>
>
> **Q4. Missing some reference about multi-view synthesis.**
>
> We have included the missing articles in our new PDF (lines 557-558, 734-736).

---

### Official Review · Reviewer_P3fm · 2024-11-04

**Soundness:** 3
**Presentation:** 3
**Contribution:** 3
**Rating:** 6
**Confidence:** 4

**Summary:**

This paper aims to synthesize 3D stereoscopic videos using an off-the-shelf monocular video generation model without finetuning. Concretely, the authors leverage the estimated video depth and propose a novel frame matrix to denoise both spatially and temporally, and thus the model is aware of the exsiting information on the left view and synthesizes consistent right-view videos accordingly. A disocclusion boundary re-injection scheme is also proposed to solve the boundary problem.

**Strengths:**

1. The proposed method is pose-free and training-free.

2. This paper is clearly written and easy to understand.

3. Extensive experiments demonstrate the effectiveness of the proposed frame matrix and the disocclusion boundary re-injection scheme.

**Weaknesses:**

1. As the model denoises along both temporal and spatial dimensions, one experiment that is missing is the investigation of varying the number of cameras between the left and right views. How does this variation impact the final quality and overall efficiency of the process? Is it feasible to use fewer internal camera views to save time?
2. Currently all the experiments have been conducted on the synthesized videos. It would be beneficial to explore how the results look like when applied to real-world videos.
3. The model heavily depends on a pre-trained depth estimation model, which can overlook thin structures and sometimes produce inaccurate results.

**Questions:**

1. Most videos feature a single movable object or minor movements. How does the proposed method perform when applied to more complex scenes, such as those with multiple moving objects and significant motions?
2. Typos:

L225 Denosing -> Denoising

L266 refence → reference

---

> ### Author Response · Authors · 2024-11-23
>
> **Q1. The investigation of varying the number of cameras between the left and right views.**
>
> Thanks for this suggestion. We try to reduce the number of cameras between left and right views, and show results in Fig. 19. Artifacts appear when the number of cameras is less than four, as fewer cameras can lead to rapid changes between video frames, which exceeds the capabilities of current video generation models.
>
>
> **Q2. Results on more complex scenes (multiple moving objects and significant motions) and real-world videos.**
>
> 1）Our method is applicable to real-world complex videos. In Fig. 23, we show results on two real-world videos, including fast-moving objects (‘train’) and multiple objects (‘dolphins’).
>
> 2） Another example with significant motion can be found in the supplementary video (i.e., ‘car drifting’).
>
>
> **Q3. The model heavily depends on a pre-trained depth estimation model, which can overlook thin structures and sometimes produce inaccurate results.**
>
> Yes, the current depth estimation model is weak at estimating thin structures; we have discussed this in our limitation part. Considering that depth estimation is fundamental to many tasks and has attracted much attention, we expect performance to improve in the near future.
>
> **Q4. Typos.**
>
> Thanks for pointing this out, we have corrected typos in the new version.

---

> > ### Comment · Reviewer_P3fm · 2024-11-26
> >
> > Thanks for the response. In terms of Q2, I saw clear black boundaries in Fig. 23. Are these the final output of the model?

---

> > > ### Author Response · Authors · 2024-11-27
> > >
> > > Thank you for your valuable feedback. We have updated our results in Fig. 23, where the boundary artifacts have been resolved. Through our investigation, we found that these artifacts were caused by inaccuracies in depth estimation. The depth estimation model we used can produce errors near object boundaries, particularly in complex scenes. Such inaccuracies can result in foreground pixels leaking into the background during the warping process, which can, in turn, cause boundary artifacts during denoising.
> > >
> > > To address this issue, we removed pixels with unreliable depth values by eroding a few boundary pixels. We believe that as depth estimation models improve in accuracy and as more refined depth post-processing strategy are adopted, this problem can be further mitigated, especially for complex scenes.

---

### Author Response · Authors · 2024-11-23

Thanks reviewers and ACs for their time and efforts.

Our rebuttal materials include official comments and a revised PDF. In the new PDF, we have made the following changes, which are highlighted in blue for easy distinction from the previous content.

- In Table 2, we added a new metric FVD.

- In Fig. 18, we incorporated our boundary re-injection design into the existing image inpainting method.

- In Fig. 19, we studied the number of cameras used between the left and right views.

- in Fig. 20 and Fig. 21, we included comparisons with training-based single-image view synthesis methods.

- In Fig. 22, we displayed diverse outputs of our method.

- In Fig. 23, we conducted experiments on complex real-world videos, including fast-moving objects as well as multiple moving objects.

- We corrected several typos and included the missing references

---

### Meta-Review · Area_Chair_xU3v · 2024-12-20

**Metareview:**

Summary: This paper presents a method for synthesizing 3D stereoscopic videos using a pre-trained monocular video generation model. The basic idea is to create a warped version of a video using estimated video depth and inpaint it using a video generation model. The core contribution is a novel frame matrix to denoise both spatially and temporally and, therefore, achieve coherent stereoscopic video generation.

Strength:
- Pose-free and training-free method.
- Extensive experimental validation (including videos from multiple models, e.g., Sora, Lumiere, WALT, and Zeroscope)

Weakness:
- Most videos tested are not complex, e.g., only single movable objects.
- There are artifacts at the disocclusion boundary and temporal inconsistency.
- Insufficient evaluation metrics

Justification:
- All three reviewers are positive about this paper. This work fills a gap in the existing literature on generating 3D stereoscopic videos. The method of using depth-warped video and inpainting is not new. However, using the proposed frame matrix in the video noising framework is interesting and novel. Most of the raised concerns by reviewers have been resolved in the rebuttal stage. The AC thus recommends to accept.

**Additional Comments On Reviewer Discussion:**

Below are the main raised concerns by the reviewers and the authors' response. Overall, the reviewers found the responses satisfactory (except for the contribution of boundary re-injection by Reviewer dHEi). The authors will moderately reduce the discussion on boundary re-injection and focus on analyzing the core contributions of the frame matrix in the final version.

Reviewer zJCL:
- disocclusion boundary artifacts

Authors: the proposed boundary reinjection method can mitigate artifacts. But this problem remains unsolved and can be improved with more advanced video generation models.

Reviewer zJCL:
- slightly lower temporal consistency compared to Deep3D

Authors: Deep3D does not provide sufficient 3D effects (as visualized in Figure 9).

Reviewer dHEi:
- analysis for denoising frame matrix

Authors: provided a detailed description on high-level reason and more analysis of frame matrix

Reviewer dHEi:
- insufficient evaluation metrics

Authors: added FVD in Table 2

Reviewer dHEi:
- lack of comparisons with related methods.

Author: included comparisons with SVG and AdaMPI.

Reviewer P3fm:
- missing results of varying the number of cameras between the left and right views

Authors: Included new results in Fig. 19

Reviewer P3fm:
- results on real videos?

Authors: The method is applicable to real videos. Fig. 23 shows two real-world videos.

---

### Decision · Program_Chairs · 2025-01-22

Accept (Poster)